# Accelerating Multimodal Large Language Models via Dynamic Visual-Token Exit and Empirical Findings

**Qiong Wu**[12], **Wenhao Lin**[12], **Yiyi Zhou**[12],[*] **Weihao Ye**[1], **Zhanpeng Zeng**[1],
**Xiaoshuai Sun**[12], **Rongrong Ji**[12]

[1] Key Laboratory of Multimedia Trusted Perception and Efficient Computing,
Ministry of Education of China, Xiamen University, 361005, P.R. China.
[2] Institute of Artificial Intelligence, Xiamen University, 361005, P.R. China.
{qiong, wenhaolin}@stu.xmu.edu.cn, zhouyiyi@xmu.edu.cn,
weihaoye@stu.xmu.edu.cn, {zzeng, xssun, rrji}@xmu.edu.cn

## Abstract

In this paper, we study the visual redundancy problem of *multimodal large language models* (MLLMs) from the perspective of attention behaviors. Via extensive empirical experiments, we observe and conclude three main inference stages of MLLMs: *(i) Early fusion* between tokens is first accomplished quickly. *(ii) Intra-modality modeling* then comes to play. *(iii) Multimodal reasoning* resumes and lasts until the end of inference. In particular, we reveal that visual tokens will stop contributing to reasoning when the text tokens receive enough image information. Based on this observation, we propose an effective method to improve the efficiency of MLLMs, termed *dynamic visual-token exit* (DyVTE), which is orthogonal but collaborative to previous token-wise visual compression methods. To validate the efficacy of DyVTE, we apply it to a set of MLLMs, including LLaVA, VILA, EAGLE and InternVL. The experimental results not only show the effectiveness of our DyVTE in improving MLLMs' efficiency, *e.g.*, DyVTE reduces the computation overhead of LLaVA-1.5 by up to 45.7% without performance drop, but also reveal a general pattern across multiple MLLMs, well facilitating the in-depth analysis of MLLMs. Our code is released at `https://github.com/DoubtedSteam/DyVTE`.

## 1 Introduction

Recently, the rapid development of vision-language learning has been witnessed with the great success of *large language models* (LLMs) [2, 6, 16, 46, 52, 53, 54]. Numerous efforts are devoted to equip LLMs with multimodal capability, *i.e.*, *multimodal large language models* (MLLMs) [11, 14, 29, 31, 36, 40, 56]. To overcome visual shortcoming [34, 49] in extreme high computation overhead, recent MLLMs often resort to higher-resolution images as input, which is often accompanied with a multitude of visual tokens [11, 15, 29, 43].

However, the multitude of visual tokens lead to prohibitively expensive computation. For instance, compared with LLaVA 1.5 [40], LLaVA-NeXT [29] adopts about 1728 more visual tokens, resulting in about 4 times more FLOPs. Despite the performance gains, recent works[62, 7, 37] also show that these large amount of tokens are obviously redundant, leading to a great waste of computation. For instance, Ye *et al.* [62] show that randomly dropping half visual tokens barely affects performance on common VL tasks, such as MMB [41] and SQA [42]. In this case, the efficient use of visual tokens has recently become a research hot-spot, and attracts an influx of attention [8, 13, 55, 57]. However,

---

[*]Corresponding Author.

39th Conference on Neural Information Processing Systems (NeurIPS 2025).

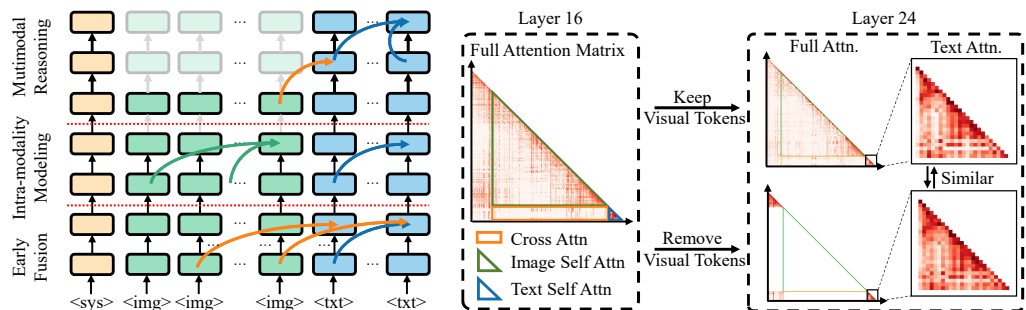

Figure 1: **Left**: Illustration of three main stages observed in MLLMs. During inference, an MLLM will go through three main stages, *i.e.*, *early fusion* between all tokens, *intra modality modeling* of the same-modal tokens, and *multimodal reasoning* between all tokens again. **Right**: The impact of visual tokens on the text self attention at the multimodal reasoning stage. Removing visual tokens at an appropriate time barely changes the text attention patterns, indicating that visual tokens contribute little to multi-modal reasoning.

existing research mainly focuses on evaluating token-wise redundancy, *e.g.*, token pruning [7, 62] and token merging [32, 47], lacking in-depth exploration of the intrinsic behaviors of MLLMs.

In this paper, we are dedicated to understand how MLLMs use visual tokens and how they behave during multimodal reasoning. To approach this target, we conduct extensive empirical studies about the attention behaviors of MLLMs, and observe and summarize three main stages of their inference, as illustrated in Fig.1-Left. At the first stage, an MLLM quickly accomplishes the exchange of multimodal information at its shallow layers, which we term *(i) early fusion*. Afterwards, this information exchange decreases, and the tokens primarily engage in intra-modal interactions, termed *(ii) intra-modality modeling*. At the last stage, visual tokens will resume the information propagation to the text ones, and this process will continue until a certain layer, and we term it *(iii) multimodal reasoning*. Through extensive comparisons, we reveal that these observed behaviors of MLLMs are common across different models [12, 36, 40, 48] and tasks [22, 41, 63].

Behind these shared behaviors of MLLMs, we observe a key finding in terms of visual redundancy, that is *visual tokens will continue to propagate their semantics to the text ones at the multimodal reasoning stage, while their impact can be very limited.* As shown in the right sub-figure of Fig.1, removing visual tokens at a suitable layer of MLLMs does not produce significant changes to the text self-attention distributions. This case suggests that visual tokens actually contribute little to multimodal reasoning at the last inference stage. In other words, multimodal reasoning only happens within text tokens after they receive enough visual semantics. This finding is also confirmed in some very recent works [37], where the manual removal of visual tokens does not decline the performance of MLLMs on some tasks. But in this paper, we also found that the optimal time to remove visual tokens is distinct for different examples, tasks and even MLLMs. And the manual exploration is experimentally expensive and hard to meet the global optimum. Therefore, *when and how to automatically remove visual tokens* is still a thorny challenge.

Motivated by these observations, we propose a novel and effective method to improve MLLMs' efficiency, termed **dynamic visual-token exit** (DyVTE). In particular, we apply lightweight hyper-networks to perceive text token status and then dynamically decide the exit of all visual tokens, thereby speeding up inference. As discussed above, DyVTE mainly focuses on the overall contribution of all visual tokens during multimodal inference, which is orthogonal but collaborative to token-wise solutions like *token pruning* [7, 27, 62]. In our experiments, we show that these two paradigms can be easily combined to boost the efficiency of MLLMs. In addition, DyVTE also differs from previous *dynamic early exiting* methods [19, 21, 51], which often refer to an inference break *i.e.*, skipping rest layers for direct predictions. In contrast, text tokens continue to transform in DyVTE, which better meets our empirical findings. Moreover, we also introduce an efficient training regime for DyVTE independent to the global gradient computation of MLLMs, achieving effective and dynamic visual-token exit with very cheap training expenditure.

To validate DyVTE, we apply it to a set of advanced MLLMs with varying scales, including LLaVA-1.5 [40], VILA [36], EAGLE [48] and InternVL [12], and conduct experiments on a bunch of widely-used VL and MLLM benchmarks [20, 22, 23, 34, 49, 63]. The experimental results show

that our DyVTE can greatly reduce the computation overhead of MLLMs, while retaining their competitive performance on various benchmarks. For instance, our DyVTE reduces the computation overhead of LLaVA-1.5 by up to 45.7% without performance drop. When combined with token pruning methods like FastV [7], DyVTE can help LLaVA-1.5 reduce up to 51.5% computation with only 0.7% performance drop on average.

Overall, our contributions are three-fold:

- We study the problem of visual redundancy from the perspective of MLLMs' behaviors, and reveal the dependency between text and visual tokens.

- Based on the empirical findings, we propose a novel and effective approach to reduce visual redundancy of MLLMs, termed *dynamic visual-token exit* (DyVTE), which can dynamically evaluate and schedule the contributions of visual tokens to multimodal reasoning.

- The extensive experiments on a set of MLLMs well validate the motivation and effectiveness of DyVTE, also providing insights into the principle of MLLMs.

## 2 Related Work

Based on the successful *large language models* (LLMs) [1, 16, 2, 14, 25], numerous *multimodal large language models* (MLLMs) have been recently proposed [31, 40, 3, 29, 11] and achieved remarkable progresses on various vision-language tasks [22, 20, 23, 60, 33]. In terms of methodology, most MLLMs often project the extracted visual features onto the semantic space of LLM for both multimodal fusion and reasoning [40, 56, 48, 12]. For instance, LLaVA [40] uses a projection layer to transform visual features into input tokens for LLaMA [54]. BLIP-2 [31] introduces *QFormer* to aggregate query-related visual features as the input visual tokens. Similarly, QWen-VL also adopts learnable tokens as queries to obtain a limited number of visual tokens [3]. To improve the ability in fine-grained tasks [49, 44, 45], some recent MLLMs resort to increasing the resolution of input images [29, 56, 10] for better visual understanding. For instance, LLaVA-NeXT [39] concatenates the tokens of each crop of the image as the input of the LLM. InternVL2.5 [10] and Qwen2-VL [4] introduce dynamic resolution designs to match different images and preserve their visual details. Despite success, these high-resolution settings also inevitably increase the number of visual tokens, leading to excessive computation overhead [7, 62].

Moreover, recent studies also show that the excessive use of visual tokens brings in obvious redundancy in both information and computation [35, 58, 59]. In this case, the efficiency study of MLLMs also becomes a research hotspot, of which focus ranges from structure design [35, 58, 59], token pruning [27, 18, 5] and token merging [47, 5]. The research of structure design mainly aim to build a new and lightweight MLLMs [35, 65], and our work is closer to the token efficiency researches [62, 7]. For instance, FastV [7] applies the averaged attention scores to evaluate the importance of each visual token and then drop the less important ones to reduce computation. FitPrune [62] resort to a statistic principle to quickly produces a token pruning regime for MLLMs based on the metrics of visual and cross attentions. PruMerge [47] speeds up the inference of MLLMs by merging visual tokens that have similar semantics. G-Prune [26] proposes an graph-based algorithm to select the most representative visual tokens to avoid redundant semantics and computations. Compared with these token-wise approaches, our DyVTE focuses more on the overall contributions of visual tokens in MLLMs, providing an alternative but orthogonal way for efficient MLLMs. In terms of dynamic inference, our works are also related to the dynamic early-exit studies for LLMs [17, 9]. However, these methods focus on the skipping of redundant layers for early prediction akin to previous dynamic methods [51, 21]. Recently, LLaVA-Mini [64] introduces a pre-fusion module through full pre-training and SFT, reducing visual tokens to just one. And our DyVTE is to conduct the early removal of visual tokens while keeping the original inference pattern. Moreover, this paper also involves quantitative analyses about the attention behaviors of MLLMs, providing insights into the mechanisms for their multimodal reasoning.

## 3 Inference Pattern Analysis of MLLMs

We first quantitatively investigate the attention behaviors of MLLMs, and then examine the impacts of visual tokens during multimodal inference.

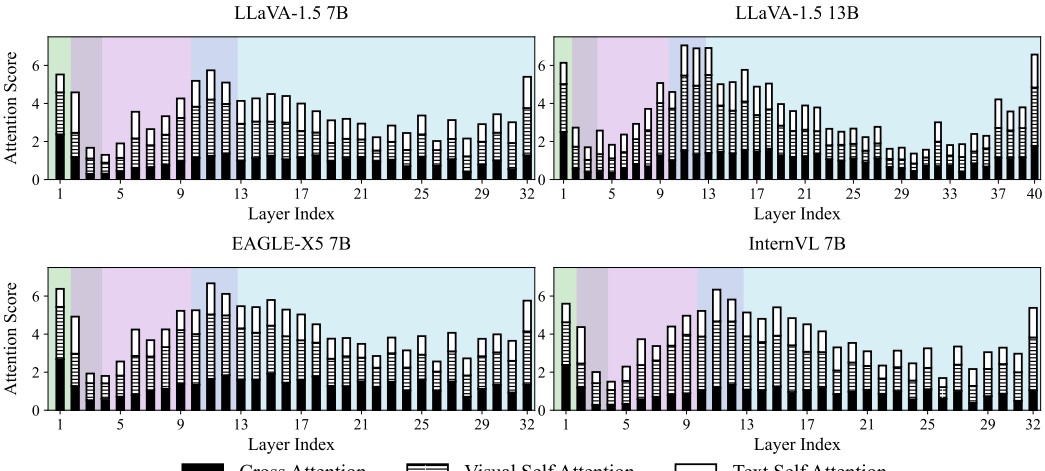

Figure 2: The averaged attention scores of four MLLMs in terms of cross, visual and text attentions. The background colors denote the three summarized stages of MLLMs, *i.e.*, *early fusion* (green), *intra-modality modeling* (purple) and multimodal reasoning (blue).

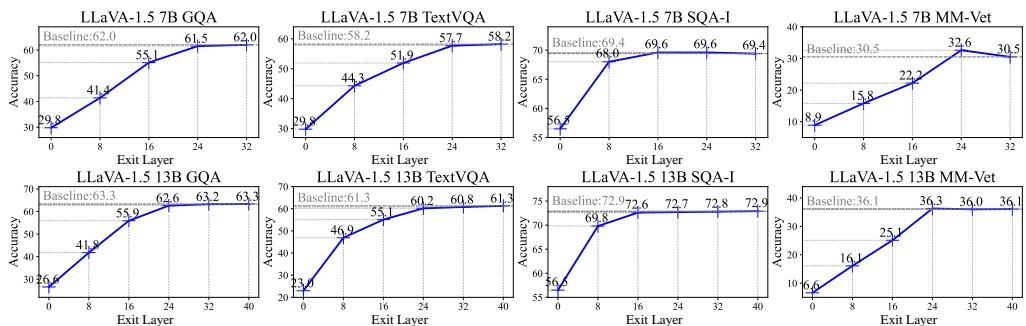

Figure 3: The relationship between manual visual-token exit and performance on LLaVA-1.5 7B and 13B. We show the results of removing all visual tokens at 0-th, 8-th, 16-th, 24-th and 32-th layers. "*Baseline*" represents the original performance of the MLLM. After a certain layer, the removal of all visual tokens barely impedes performance, but the layers for removal are different for different tasks.

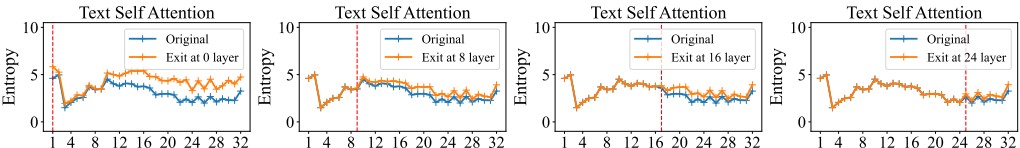

Figure 4: The entropy of text self-attention distribution with different layer to remove all visual tokens. We visualize the entropy of text self-attention on LLaVA-1.5 7B on GQA benchmark with different with different layer to remove all visual tokens, which shows that the removal of visual tokens after a certain layer barely affect the text self-attention.

**Attention behaviors of MLLMs.** We first visualize the attention patterns of four representative MLLMs [40, 48, 12] of two scales in Fig.2. These attention data includes *visual self-attention*, *visual-text cross-attention* and *text self-attention*[2]. Here, the values in these distributions are the column-wise summarized attention scores on the *LLaVA-split* [40]. From these plots, we can first observe that four MLLMs share similar patterns for three types of attentions. At the shallow layers, all MLLMs show high cross-attention values between visual and text tokens, suggesting the intensive interactions between two modalities. This corresponds to the *early fusion* stage we termed above, and it denotes the quick exchange of multimodal information. But *early fusion* declines quickly, and we can observe that the self-attention between the tokens of the same modality becomes the main activity at the second stage, *i.e.*, the modeling mainly involved in each modality. In this case, we can assume that MLLM starts the *intra-modality modeling* stage. At the last stage, the visual tokens will

---

[2]The visualization details are provided in our appendix A.1.

resume information propagation to the text tokens, and this process will continue to oscillate until the end of inference. Here, we term this process as the *multimodal reasoning* stage. Notably, we can find these patterns exist in MLLMs of different families and scales.

**The impact of dropping all visual tokens.** Next, we quantitatively measure the importance of visual tokens. In Fig.3, we present the results of removing visual tokens of LLaVA-1.5 7B and 13B on four benchmarks at different layers. From these plots, we can first observe that visual tokens are not always required throughout the entire process, especially the last stage. For instance, on GQA, removing all visual tokens at 24-th layer only has $0.5\%$ and $0.6\%$ performance drops for LLaVA-1.5 7B and 13B, receptively. This finding is also consistent with some recent efforts [37] that visual tokens receive little attention in deep layers. Another finding is that the removal of visual tokens have little impact on the modeling among text tokens. As shown in Fig.4, the removal of visual tokens that does not harm the performance and also has no impact on the modeling pattern among the text tokens, *e.g.*, after $24^{th}$ layer on GQA. The only difference lies in the slight change of attention intensity due to the shorter token sequence. More importantly, we can find that *the exit time of visual tokens is different for different examples and tasks.* For instance, the optimal exit layer of SQA is earlier than that of TextVQA, *i.e.*, 16 *v.s.* 25, suggesting the need of dynamic early-exist modeling. Overall, these results confirm our argument that visual tokens stop contributing to prediction after certain layer, and also shows the need of exploring dynamic and automatic strategies for effective visual token removal.

**Discussion.** Based on the above observations and analyses, we can categorize the inference of MLLMs into three main stages. (i) *Early Fusion*. In the initial stage, the MLLM quickly accomplish the exchange of multimodal information from visual to text tokens. (ii) *Intra-modality Modeling*. Next, the self-attention intra the modality becomes the main activity, enhancing vision and language understanding. (iii) *Multimodal Reasoning*. Finally, the visual tokens will resume the information propagation to the text tokens, though this transfer demonstrates limited impact on final response generation. In addition to these three stages, we can also conclude one common behavior of visual tokens in MLLMs, *i.e.*, the contribution of visual tokens to multimodal prediction diminish after some layers. Moreover, these results also show that the exist times for different types of VL examples are different, well motivating our exploration of dynamic visual-token exit.

# 4 Dynamic Visual-token Exiting

## 4.1 Method

In this paper, we propose a novel and effective method to reduce the visual redundancy of MLLMs, termed *dynamic visual-token exit* (DyVTE). DyVTE uses lightweight hyper-networks to perceive the text token status of MLLMs and then adaptively judge the right time to remove all visual tokens.

Concretely, given the text tokens $\mathbf{T}^{(k)}$ at the $k$-th layer of an MLLM, we use a simple MLP as the hyper-network to learn their status and decide whether to remove all visual tokens, defined by

$$\mathbf{p} = Softmax\big(\text{GELU}([avg(\mathbf{T}_{1:t-1}^{(k)}), \mathbf{T}_t^{(k)}]\mathbf{W}_1)\mathbf{W}_2\big), \tag{1}$$

where $\mathbf{p} \in \mathcal{R}^2$ is the binary prediction, $\mathbf{W}_1 \in \mathcal{R}^{2d \times h}$ and $\mathbf{W}_2 \in \mathcal{R}^{h \times 2}$ are weight matrices, and $h$ is the dimension of the hidden states.

In particular, the hyper-network of Eq.1 uses two text token representations, *i.e.*, $avg(\mathbf{T}_{1:t-1})$ and $\mathbf{T}_t$. The former is the averaged text tokens representing their overall state. The latter $\mathbf{T}_t$ is the last token, which often plays an important role for decoding under the *uni-directional self-attention* setting of MLLMs [38, 50]. With these two types of text representations, DyVTE can well perceive the state of MLLMs and automatically decide whether additional visual information is still needed for reasoning.

When the prediction $\mathbf{p}$ is *exit*, DyVTE will remove all visual tokens after this layer, while the text ones are kept in the rest layers of MLLMs. This process can be denoted by

$$P_l' = G_{l+1:L}(\mathbf{T}^{(l)}), \tag{2}$$

where $G_{l+1:L}(\cdot)$ denote the remaining layers in the MLLM. Compared with previous dynamic exit methods that skip layers for early prediction [17, 9], the principle of our DyVTE is to make the removal of visual tokens based on the text token status. Thus, its implementation requires no structure tweaks, *e.g.*, adding new prediction layers, and also no the update of MLLMs.

## 4.2 Optimization

In particular, given sufficient examples of visual-token exit, DyVTE can learn to judge the text token status at each layer of the MLLM. Thus, its accurate removals of visual tokens will greatly speed up the inference in the rest layers of the MLLM, while retaining similar predictions.

Thus the objective of DyVTE can be defined by

$$\underset{\theta_h}{\arg\min}\, d(P, P'_l), \tag{3}$$

where $\theta_h$ denotes the weights of hyper-networks in DyVTE. $d(\cdot, \cdot)$ is KL-Divergence [28]. And $P'_l$ and $P$ are the predictions of an MLLM with and without early visual token exit, respectively.

To optimize Eq.3, a direct approach is to merge the predictions of hyper-networks with visual tokens of MLLMs, *e.g.,* implementing attention masks based on *Gumbel softmax* [24], thus using the *next token prediction* loss to indirectly update DyVTE. Although feasible, this approach still requires the computation of the model's all gradients, which is still inefficient and expensive.

In this case, we propose an efficient training regime independent to the gradient back-propagation of MLLMs. Concretely, we can compare the discrete outputs of the MLLM with and without DyVTE, *e.g.*, the answer strings $A$. If the answers are exactly the same, we can give a positive feedback to hyper-networks, and *vice verse*. By comparing it with the default output $A$, we can judge that whether the visual tokens should be exited at $l$-th layer:

$$\mathbf{y} = \begin{cases} 1, & A'_l = A, \\ 0, & A'_l \neq A. \end{cases} \tag{4}$$

Here, $A'_l$ denotes the answer predicted with visual-token exit at the $l$-th layer. Besides, to make this supervision more robust, we also consider the prediction uncertainty as a regularization term, thereby , making the MLLM behaviors closer to the default inference:

$$\mathbf{y} = \begin{cases} 1, & A'_l = A \wedge \rho'_c < \tau, \\ 0, & otherwise. \end{cases} \tag{5}$$

Here, $\rho'_c$ denotes the prediction uncertainty valued by *cross-entropy* and multiplied by a scaling factor, and $\tau$ denotes the threshold.

With this supervision, the hyper-networks in DyVTE can be optimized by the cross-entropy loss:

$$\mathcal{L}_D = -\big(\mathbf{y} \cdot \log(\mathbf{p}_1) + (1 - \mathbf{y}) \cdot \log(\mathbf{p}_0)\big). \tag{6}$$

Compared with the indirect optimization using *next token prediction* [61], Eq.5 can provide more effective supervisions to DyVTE. We randomly sample exit layers at each training step, and compute the labels $\mathbf{y}$ according to Eq.5. Finally, the hyper-networks are optimized based loss defined in Eq.6. The overall expenditure of DyVTE training will be much cheaper than tuning MLLMs or learning new prediction layers in previous dynamic exit approaches [9, 61].

# 5 Experiment

## 5.1 Datasets and Metrics

The benchmarks used in this paper consist of four conventional vision-language (VL) benchmarks and five newly introduced MLLM benchmarks. The traditional VL benchmarks include VQAv2 [22], GQA [23], ScienceQA [42], and TextVQA [49]. The MLLM benchmarks comprise POPE [34], MME [20], MMB [41], SEED [30] and MM-Vet [63]. Different from general VL evaluation, the MLLM benchmarks focus more on the evaluations of MLLMs, such as *fine-grained reasoning* [63] and *visual hallucination* [34].

## 5.2 Implementation Details

We apply DyVTE to five popular MLLMs, namely EAGLE-X5 7B [48], VILA 7B [36], InternVL 7B [12], LLaVA-1.5 7B [40] and LLaVA-1.5 13B [40]. The scale for $\rho'$ in Eq.5 is set to $1.03$. For all MLLMs, the hidden dimension of the hyper-network is set to 2,048. During training, all MLLMs are frozen, and the training rate of hyper-networks is set to $4 \times 10^{-5}$. And we randomly sample 1% examples of the *LLaVA-665k* instruction set [40] to train DyVTE for 1 epoch. More details can refer to our code project.

Table 1: Results of MLLMs with and without DyVTE on five MLLM benchmarks. The accuracy (higher is better) and TFLOPs (lower is better) are reported. The relative percentage change from the baseline model to DyVTE is also shown in parentheses.

| Method | SEED | | MME | | MMB | | POPE | | MM-Vet | |
|---|---|---|---|---|---|---|---|---|---|---|
| | Accuracy ↑ | TFLOPs ↓ | Score ↑ | TFLOPs ↓ | Accuracy ↑ | TFLOPs ↓ | Accuracy ↑ | TFLOPs ↓ | Accuracy ↑ | TFLOPs ↓ |
| EAGLE-X5 7B | 73.9 | 47.8 | 1528.0 | 27.8 | 68.4 | 29.6 | 88.8 | 27.7 | 37.4 | 27.6 |
| EAGLE-DyVTE 7B | 73.6 (-0.4%) | 43.0 (-10.0%) | 1581.7 (+3.5%) | 20.3 (-27.0%) | 68.8 (+0.6%) | 23.7 (-19.9%) | 88.4 (-0.5%) | 20.0 (-27.8%) | 37.8 (+1.1%) | 23.5 (-14.9%) |
| VILA 7B | 61.7 | 9.2 | 1489.2 | 8.9 | 69.9 | 9.5 | 86.3 | 8.8 | 36.3 | 8.7 |
| VILA-DyVTE 7B | 61.8 (+0.2%) | 5.9 (-35.9%) | 1503.1 (+0.1%) | 4.6 (-48.3%) | 69.8 (-0.1%) | 6.0 (-36.8%) | 85.6 (-0.8%) | 4.5 (-48.9%) | 36.7 (+1.1%) | 6.6 (-24.1%) |
| InternVL 7B | 59.2 | 16.0 | 1525.1 | 15.5 | 64.6 | 16.2 | 86.4 | 15.4 | 31.2 | 15.4 |
| InternVL-DyVTE 7B | 59.1 (-0.2%) | 11.9 (-25.6%) | 1474.1 (-3.3%) | 10.9 (-29.7%) | 64.4 (-0.3%) | 12.0 (-25.9%) | 81.3 (-5.9%) | 10.9 (-29.2%) | 29.5 (-5.4%) | 13.0 (-15.6%) |
| LLaVA-1.5 7B | 58.6 | 9.2 | 1510.7 | 8.9 | 64.3 | 9.6 | 85.9 | 8.8 | 30.5 | 8.7 |
| LLaVA-DyVTE 7B | 58.6 (0.0%) | 5.0 (-45.7%) | 1491.4 (-1.3%) | 4.3 (-51.7%) | 64.7 (+0.6%) | 5.4 (-43.8%) | 81.6 (-5.0%) | 4.1 (-53.4%) | 31.9 (+4.6%) | 6.3 (-27.6%) |
| LLaVA-1.5 13B | 61.6 | 17.6 | 1531.3 | 16.9 | 67.7 | 18.3 | 85.9 | 16.8 | 36.1 | 16.7 |
| LLaVA-DyVTE 13B | 59.3 (-3.7%) | 7.1 (-59.7%) | 1546.4 (+1.0%) | 7.2 (-57.4%) | 66.0 (-2.5%) | 7.8 (-57.4%) | 84.8 (-1.3%) | 7.6 (-54.8%) | 34.8 (-3.6%) | 10.6 (-36.5%) |

Table 2: Results of MLLMs with and without DyVTE on four VL benchmarks. The accuracy (higher is better) and TFLOPs (lower is better) are reported. The relative percentage change from the baseline model to DyVTE is also shown in parentheses.

| Method | GQA | | VQA | | TextVQA | | SQA-I | | Average | |
|---|---|---|---|---|---|---|---|---|---|---|
| | Accuracy ↑ | TFLOPs ↓ | Score ↑ | TFLOPs ↓ | Accuracy ↑ | TFLOPs ↓ | Accuracy ↑ | TFLOPs ↓ | Accuracy ↑ | TFLOPs ↓ |
| EAGLE-X5 7B | 64.9 | 27.8 | 83.4 | 27.8 | 71.2 | 29.5 | 69.8 | 29.2 | 72.3 | 28.6 |
| EAGLE-DyVTE 7B | 62.4 (-3.9%) | 21.7 (-21.9%) | 82.6 (-1.0%) | 21.6 (-22.3%) | 70.2 (-1.4%) | 24.5 (-16.8%) | 71.7 (+2.7%) | 23.5 (-24.3%) | 71.7 (-0.8%) | 22.8 (-20.3%) |
| VILA 7B | 63.1 | 8.8 | 80.3 | 8.8 | 62.6 | 9.5 | 69.5 | 9.8 | 68.8 | 9.2 |
| VILA-DyVTE 7B | 61.9 (-1.9%) | 5.5 (-37.5%) | 79.2 (-1.4%) | 5.4 (-38.6%) | 61.2 (-2.2%) | 7.2 (-24.2%) | 69.5 (0.0%) | 6.1 (-37.8%) | 67.9 (-1.3%) | 6.0 (-34.8%) |
| InternVL 7B | 62.9 | 15.4 | 79.3 | 15.4 | 57.0 | 16.1 | 66.2 | 16.4 | 66.4 | 15.8 |
| InternVL-DyVTE 7B | 61.3 (-2.5%) | 11.8 (-23.4%) | 77.6 (-2.1%) | 11.7 (-24.0%) | 55.8 (-2.1%) | 13.5 (-16.1%) | 66.2 (0.0%) | 12.1 (-26.2%) | 65.2 (-1.8%) | 12.3 (-22.2%) |
| LLaVA-1.5 7B | 62.0 | 8.8 | 78.5 | 8.8 | 58.2 | 9.5 | 69.4 | 9.8 | 67.0 | 9.2 |
| LLaVA-DyVTE 7B | 60.0 (-3.2%) | 5.3 (-39.8%) | 76.6 (-2.4%) | 5.1 (-42.0%) | 56.6 (-2.7%) | 6.7 (-29.5%) | 69.6 (+0.3%) | 5.5 (-43.9%) | 65.7 (-1.9%) | 5.6 (-39.1%) |
| LLaVA-1.5 13B | 63.3 | 16.8 | 80.0 | 16.8 | 61.3 | 18.1 | 72.9 | 18.6 | 69.4 | 17.6 |
| LLaVA-DyVTE 13B | 62.3 (-1.6%) | 9.0 (-46.4%) | 78.8 (-1.5%) | 8.9 (-47.0%) | 58.9 (-3.9%) | 10.8 (-40.3%) | 72.3 (-0.8%) | 8.2 (-55.9%) | 68.1 (-1.9%) | 9.2 (-47.7%) |

## 5.3 Quantitative Analysis

**Results of DyVTE on different MLLMs.** In Tab.1 and Tab.2, we report the results of applying DyVTE to a set of MLLMs of different families and sizes, including EAGLE [48], VILA [36], InternVL [12], and LLaVA-1.5 [40]. From these tables, we can first observe that DyVTE significantly reduces the computational overhead of existing MLLMs. For example, DyVTE reduces FLOPs of LLaVA 7B by 45.7% on SEED without dropping performance. Additionally, the actual reduction of FLOPs are distinct for different tasks. For instance, on SQA with multiple-choice questions, DyVTE reduces computational overhead by up to 43.9%, and on MM-Vet about granular answers, the reduction achieved is 27.6%. These results confirm our intuition that the optimal removal of visual tokens are different for different tasks. Another observation is that DyVTE's effects vary across these MLLMs. Specifically, DyVTE can reduce the computational overhead of VILA 7B by 48.3% on the MME benchmark with no performance drop, whereas InternVL-DyVTE 7B only achieve a reduction of 29.7%. When applying DyVTE to larger MLLMs, such as LLaVA-1.5 13B, we can observe a more significant efficiency improvement. For instance, DyVTE reduces the computational overhead of LLaVA-1.5 13B by 55.9% on SQA with only -0.8% on performance. Furthermore, DyVTE is more effective for MLLMs that uses stronger visual representations. For instance, EAGLE-X5 is a SOTA MLLM with strong and hybrid visual backbones, and it works very well with our DyVTE. DyVTE can reduce its computation by up to 6.2 TFLOPs with only 0.8% performance drop on the VL benchmarks. Overall, these results not only validate the effectiveness of our DyVTE but also confirm our motivations and findings about MLLMs.

Table 3: Comparison of real inference budget between token pruning methods and vanilla decoding. "*Acc.*" denotes the accuracy.

| Method | SQA-I | | MMB | |
|---|---|---|---|---|
| | Latency ↓ | Acc. ↑ | Latency ↓ | Acc. ↑ |
| LLaVA 13B | 0.237s | 72.9 | 0.236s | 67.7 |
| FastV | 0.171s (-27.7%) | 73.1 (+0.3%) | 0.174s (-26.2%) | 68.6 (+1.3%) |
| ToMe | 0.175s (-26.2%) | 73.2 (+0.4%) | 0.179s (-24.2%) | 67.4 (-0.4%) |
| **DyVTE** | 0.161s (-32.1%) | 72.3 (-0.8%) | 0.163s (-30.9%) | 66.0 (-2.5%) |
| **DyVTE**+FastV | 0.154s (-34.9%) | 73.0 (+0.2%) | 0.155s (-34.4%) | 68.5 (+1.2%) |

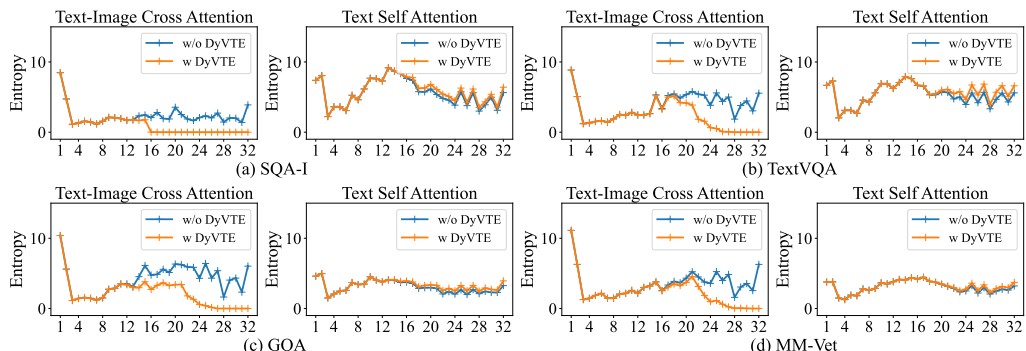

Figure 5: The entropy of two attention distributions. We visualize the entropy of cross-attention matrices and text self-attention on LLaVA-1.5 7B with and without our DyVTE, which shows that the removal of visual tokens barely affect the text self-attention.

Table 4: Ablation study of different token statuses for DyVTE. "*Mean Visual*" denotes the average of all visual tokens except the last one, similar with "*Mean Text*". "*Last Visual*" and "*Last Text*" denotes the last visual or text token. "*Exit Layer*" denotes the averaged layer numbers selected to remove by DyVTE. "*Acc.*" denotes the accuracy. The default setting of DyVTE is the last row. The best and second best results are marked in **bold** and underline, receptively.

| State | | | | GQA | | TextVQA | | MM-Vet | | SQA-I | | Average | |
|---|---|---|---|---|---|---|---|---|---|---|---|---|---|
| Mean Visual | Last Visual | Mean Text | Last Text | Acc. ↑ | Exit Layer ↓ | Acc. ↑ | Exit Layer ↓ | Acc. ↑ | Exit Layer ↓ | Acc. ↑ | Exit Layer ↓ | Acc. ↑ | Exit Layer ↓ |
| ✓ | | | | **61.4** | 21.6 | 57.3 | 21.0 | **31.5** | 21.4 | 69.7 | 21.7 | **55.0** | 21.4 |
| | ✓ | | | 61.2 | 21.3 | 57.1 | 21.1 | 30.0 | **21.0** | 69.7 | 21.4 | 54.5 | 21.2 |
| | | ✓ | | 60.1 | 17.4 | **57.6** | 22.1 | 31.1 | 22.1 | 69.7 | 20.4 | 54.6 | 20.5 |
| | | | ✓ | 59.8 | **16.3** | 56.3 | **19.1** | 29.9 | **21.0** | **69.8** | **13.8** | 54.0 | **17.6** |
| ✓ | ✓ | | | **61.1** | 21.2 | 57.3 | 21.3 | 31.6 | 21.3 | 69.7 | 21.0 | **54.9** | 21.2 |
| ✓ | | ✓ | | 58.7 | 16.5 | **57.6** | 22.2 | **32.7** | 22.5 | 69.7 | 21.3 | 54.7 | 20.6 |
| ✓ | | | ✓ | 59.4 | **16.3** | 56.5 | 19.9 | 30.9 | 21.8 | 69.7 | 14.4 | 54.1 | 18.1 |
| | ✓ | ✓ | | 59.4 | 16.7 | 57.4 | 21.8 | 31.3 | 22.0 | 69.6 | 20.3 | 54.4 | 20.2 |
| | ✓ | | ✓ | 59.3 | 16.4 | 56.5 | 19.9 | 31.2 | 21.6 | 69.6 | 14.4 | 54.1 | 18.1 |
| | | ✓ | ✓ | 60.0 | 16.4 | 56.6 | **19.7** | 31.9 | **21.1** | 69.6 | **13.4** | 54.5 | **17.6** |

**Inference Latency.** In Tab. 3, we compare the actual inference latency of DyVTE with the token prune and the vanilla decoding methods. From the table, we can first observe that token prune methods can significantly reduce the inference latency. For instance, FastV can reduce the inference time by -27.7% and -26.2% on SQA and MMB benchmarks, receptively. Compared with these token prune methods, our DyVTE further enhances inference efficiency. Specifically, DyVTE improve the inference efficiency by 32.1% and 30.9% on SQA and MMB benchmarks with competitive performance. Moreover, the proposed DyVTE can work together with existing token prune methods and achieve better efficiency. For example, when combined with FastV, DyVTE achieves even greater reductions in inference latency, *i.e.*, 34.9% and 34.4% on SQA and MMB benchmarks. Overall, these results validate our DyVTE can accelerate the inference.

**The attention entropy with visual tokens exit.** In Fig.5, we visualize the entropy distributions of cross-attention matrices and text self-attention matrices of LLaVA-1.5 7B. And we compare the results with and without DyVTE across four benchmarks. We first observe that there are differences between the multimodal reasoning procedures across different benchmarks. Specifically, in SQA and TextVQA, the changes in entropy during text modeling are more obvious than those in GQA and MM-Vet. This phenomenon fully illustrates the necessity of adopting a dynamic approach to exit visual tokens. Another observation is that DyVTE can effectively remove all visual tokens at a specific layer according to the status of multimodal reasoning. Specifically, after removing all visual tokens by DyVTE, the entropy of text self-attention remains unaffected. In this way, the exiting determined by DyVTE only blocks visual information's cross-modality propagation while preserving text tokens' modeling. Overall, these findings validate that visual tokens are not always necessary in the reasoning process, and also confirm the effectiveness of the proposed DyVTE.

**Ablation study.** In Tab.4, we perform a set of experiments to analyze the effectiveness of different token status for DyVTE. In this table, various tokens are used as the token statuses in Eq.1. As shown in Tab.4, when using only image-related tokens, the model tends to remove all visual tokens at a later stage. Specifically, the average exit layer are 21.4 and 21.2. Using the "*Mean Text*" token leads to an earlier exit from the layers, but still retains the computation redundancy. When only the "*Last Text*" token is used, DyVTE removes all visual tokens too early, *i.e.*, at the 17-th layer, resulting

Table 5: Comparison between DyVTE and token pruning methods. The best and second best results are marked in **bold** and underline.

| Method | SQA-I | | MM-Vet | | SEED | | MMB | | Average | |
|---|---|---|---|---|---|---|---|---|---|---|
| | Accuracy ↑ | TFLOPs ↓ | Score ↑ | TFLOPs ↓ | Accuracy ↑ | TFLOPs ↓ | Accuracy ↑ | TFLOPs ↓ | Accuracy ↑ | TFLOPs ↓ |
| LLaVA 7B | 69.4 | 9.8 | 30.5 | 8.7 | 58.6 | 9.2 | 64.3 | 9.6 | 55.7 | 9.3 |
| ToMe [5] | **69.6** (+0.3%) | 5.9 (-39.8%) | 30.6 (+0.3%) | 4.9 (-43.7%) | 57.8 (-1.4%) | 5.5 (-40.2%) | 63.7 (-0.9%) | 5.7 (-40.6%) | 55.4 (-0.5%) | 5.5 (-40.9%) |
| FastV [7] | 69.0 (-0.6%) | 6.2 (-36.7%) | 31.3 (+2.6%) | 5.2 (-35.8%) | 58.2 (-0.7%) | 5.8 (-37.0%) | 64.4 (+0.2%) | 6.0 (-37.5%) | 55.7 (0.0%) | 5.8 (-37.6%) |
| **DyVTE** | **69.6** (+0.3%) | 5.5 (-43.9%) | **31.9** (+4.6%) | 6.3 (-27.6%) | **58.6** (0.0%) | 5.0 (-45.7%) | **64.7** (+0.6%) | 5.4 (-43.8%) | **56.2** (+0.9%) | 5.5 (-40.9%) |
| DyVTE+FastV | 68.9 (-0.7%) | **4.8** (-51.0%) | 29.8 (-2.3%) | **4.0** (-54.0%) | 58.2 (-0.7%) | **4.6** (-50.0%) | 64.4 (+0.2%) | **4.6** (-52.1%) | 55.3 (-0.7%) | **4.5** (-51.5%) |

Table 6: Results of MLLMs with DyVTE that trained on themselves and others on five benchmarks. "*Trained*" denotes the MLLM that DyVTE trained on. "*Exit Layer*" denotes the averaged layer numbers selected to remove by DyVTE. "*Acc.*" denotes the accuracy.

| MLLM | Trained | GQA | | VQAv2 | | SEED | | MMB | | MME | |
|---|---|---|---|---|---|---|---|---|---|---|---|
| | | Acc. ↑ | Exit Layer ↓ | Acc. ↑ | Exit Layer ↓ | Acc. ↑ | Exit Layer ↓ | Acc. ↑ | Exit Layer ↓ | Acc. ↑ | Exit Layer ↓ |
| VILA 7B | - | 63.1 | - | 80.3 | - | 61.7 | - | 69.9 | - | 1489.2 | - |
| | VILA 7B | 61.9 | 17.4 | 79.2 | 16.7 | 61.8 | 17.6 | 69.8 | 16.2 | 1503.1 | 13.1 |
| | LLaVA 7B | 61.3 | 17.1 | 79.1 | 17.1 | 61.8 | 15.8 | 69.8 | 16.5 | 1504.9 | 14.6 |
| InternVL 7B | - | 62.9 | - | 79.3 | - | 59.2 | - | 64.6 | - | 1525.1 | - |
| | InternVL 7B | 61.3 | 16.1 | 77.6 | 15.8 | 59.1 | 13.9 | 64.4 | 13.6 | 1474.1 | 12.1 |
| | LLaVA 7B | 61.3 | 17.0 | 77.7 | 16.8 | 59.2 | 18.1 | 64.9 | 17.8 | 1516.9 | 14.1 |

in an obvious performance drop, *i.e.*, performance drops by 1.0% on average. For combinations of two representations, we can observe that the "*Mean Text + Last Text*" is the most effective way. Specifically, it removes visual tokens at about 17-th layer, while causing only a 0.5% performance drop. On the other hand, using a combination of visual token statuses, *i.e.*, "*Mean Visual + Last Visual*", yields results similar to using a single visual representation. In this way, the visual tokens are retained until the 21-st layer. Overall, these results confirm that the key to judging whether visual tokens have a necessary impact on the prediction lies in the status of text tokens.

**Generalization to different MLLMs.** To evaluate the generalization capability of DyVTE across different MLLMs, we conduct experiments where DyVTE is trained on one MLLM and directly applied to another without additional fine-tuning. As shown in Tab.6, we compare the performance of DyVTE trained on the same MLLM versus trained on LLaVA 7B and applied to VILA 7B and InternVL 7B. The results demonstrate that DyVTE exhibits strong cross-model generalization. For instance, when applying DyVTE trained on LLaVA to VILA, the model achieves 61.3% accuracy on GQA with an average exit layer of 17.1, which is comparable to the specifically trained DyVTE (61.9% accuracy with exit layer 17.4). Another notable observation is that the optimal exit layers identified by cross-model DyVTE are close to those of specifically tuned DyVTE. On the MME benchmark, both VILA with LLaVA-trained DyVTE (exit layer 14.6) and VILA with self-trained DyVTE (exit layer 13.1) achieve similar efficiency gains. These results indicate that DyVTE has learned generalized patterns of visual token importance across different MLLM architectures. This strong generalization capability stems from the task definition of DyVTE as a simple binary prediction problem, which makes the optimization independent of specific MLLM gradient updates and enables direct supervision via binary labels based on MLLMs' predictions. Overall, the cross-model experiments validate that DyVTE can be trained once and effectively deployed across different MLLMs with minimal performance degradation.

**Comparison with token pruning methods.** In Tab.5, we compare the performance and efficiency of DyVTE with the visual token pruning methods on LLaVA-1.5 7B. From the table, we can observe that DyVTE outperforms token pruning methods in both performance and efficiency. For example, when compared with ToMe on SQA, DyVTE not only maintains the performance but also further reduces FLOPs by 4.1%. When compared with FastV, the advantage of DyVTE is more significant. Specifically, DyVTE improves performance by 0.4% while reducing computation overhead by 8.7% on the SEED. Another observation is that DyVTE can significantly improve performance in complex tasks. For example, in the MM-Vet benchmark, which evaluates multiple functions including mathematics and OCR, DyVTE outperforms ToMe by 4.3Additionally, DyVTE shows excellent compatibility with existing methods. When combined with FastV, DyVTE reduces the computational overhead by 51.5% while only decreasing the performance by 0.7% on average. Overall, these results validate the effectiveness and efficiency of DyVTE.

## 5.4 Qualitative Analysis

To gain insight into the proposed DyVTE, we visualize the effect of applying it on LLaVA-1.5 7B in Fig. 6. We can first observe that DyVTE maintains consistent predictions with the default LLaVA. As shown in Figure 6-(a), DyVTE can correctly identify the object in question even when the foreground is complex. Additionally, DyVTE provides much faster inference than the default LLaVA. Specifically, in the two given examples, DyVTE can improve inference speed by up to 10.2% compared to the default LLaVA.

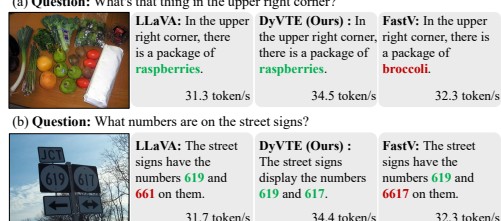

Figure 6: Examples on LLaVA-1.5 with DyVTE. Our DyVTE can help the MLLM answer the questions as accurately as default, while being faster.

Another observation is that removing redundant visual information may enhance performance. As shown in Fig.6-(b), applying DyVTE can help LLaVA to better recognize the numbers on the sign. Overall, these results confirm that DyVTE enhances inference efficiency without harming the reasoning process of the default model, aligning well with our motivation.

## 6 Limitation

While DyVTE shows promising efficiency gains across multiple MLLMs and benchmarks, DyVTE currently performs full visual-token exit, which may not be optimal for cases where partial visual information is still beneficial in later layers.

## 7 Conclusion

In this paper, we investigate the visual redundancy problem of MLLMs via analyzing their inference behaviors, and summarize three key stages of MLLMs during multimodal inference, *i.e.*, *early fusion*, *inter-modality modeling*, and *multimodal reasoning*. Moreover, we also reveal that visual tokens stop contributing to multimodal reasoning after some layers. Motivated by this insight, we propose a simple yet effective method for MLLMs, termed Dynamic Visual-Token Exit (DyVTE) which optimally determines the layer at which visual tokens can be removed, thereby reducing computational overhead without compromising performance. Extensive experiments on nine benchmarks demonstrate that DyVTE significantly enhances the efficiency of MLLMs while maintaining their performance.

## 8 Acknowledgment

This work was supported by the National Science Fund for Distinguished Young Scholars (No.62025603), the National Natural Science Foundation of China (No. U21B2037, No. U22B2051, No. U23A20383, No. U21A20472, No. 62176222, No. 62176223, No. 62176226, No. 62072386, No. 62072387, No. 62072389, No. 62002305 and No. 62272401), and the Natural Science Foundation of Fujian Province of China (No. 2021J06003, No.2022J06001).

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

# A Appendix

In this paper, we first conduct extensive empirical studies on the attention behaviors of MLLMs, and summarize three main inference stages in MLLMs: *(i) Early fusion* between tokens is first accomplished quickly. *(ii) Intra-modality modeling* then comes to play. *(iii) Multimodal reasoning* resumes and lasts until the end of inference. To better confirm our findings, we conduct further experiments on more MLLMs in the supplementary materials.

## A.1 Attention Visualization

In Fig.2, we visualize the attention weight of each part in attention matrix, and the attention weights are calculated as follow: We first calculate the attention matrix with system prompt $\mathbf{S} \in \mathcal{R}^{s \times d}$, visual tokens $\mathbf{V} \in \mathcal{R}^{v \times d}$ and text tokens $\mathbf{T} \in \mathcal{R}^{t \times d}$:

$$\mathbf{A} = Softmax(\frac{[\mathbf{S}, \mathbf{V}, \mathbf{T}]\mathbf{W}_q([\mathbf{S}, \mathbf{V}, \mathbf{T}]\mathbf{W}_k)^T}{\sqrt{d}} \cdot \mathbf{M}), \quad (7)$$

where $\mathbf{M}$ represents the attention mask for causal reasoning. Then, the weight of each part can be represent as

$$W_v = \sum_{i=s}^{s+v}\sum_{j=s}^{s+v} \mathbf{A}_{ij}/v, W_c = \sum_{i=s+v}^{s+v+t}\sum_{j=s}^{s+v} \mathbf{A}_{ij}/t, W_t = \sum_{i=s+v}^{s+v+t}\sum_{j=s+v}^{s+v+t} \mathbf{A}_{ij}/t \quad (8)$$

where $W_v$, $W_c$ and $W_t$ represent the attention weights of visual self attention, cross attention and text self attention, respectively. In addition, we combine the attention matrices of different heads in MHA through the *max pooling*.

## A.2 Ablation Study

In Tab.7 and Tab.8, we perform a set of experiments to analyze the impact of different representation selections on DyVTE. As shown in Tab.7, when relying on more than two representations, we can find out that their performance is similar to "*mean text + last text*". For instance, the additional representation, *i.e.*, "*last visual*" and "*mean visual*", has no noticeable effect on the results. We can also observe that the missing of "*mean text*" or "*last text*" will lead to inaccurate exit. For example, the missing of "*mean text*" leads to later exit, *i.e.*, 2 layers latter, while the absence of "*last text*" does the opposite, *i.e.*, 2 layers earlier. As for taking attention scores be the representations, we find that hyper-network can not make the correct judgment. Specifically, the exit layers for all benchmarks can be much earlier. Overall, these results well confirm the use of text tokens status for dynamic visual token exiting in MLLMs.

Table 7: Ablation study of the token status selection for DyVTE. "*Mean Visual*" denotes the average of all visual tokens except the last one, similar with "*Mean Text*". "*Last Visual*" and "*Last Text*" denotes the last visual or text token. "*Exit Layer*" denotes the averaged layer numbers selected to remove by DyVTE. "*Acc.*" denotes the accuracy. The default setting of DyVTE is the last row. The best and second best results are marked in **bold** and underline, receptively.

| Mean Visual | Last Visual | Mean Text | Last Text | GQA Acc. ↑ | GQA Exit Layer ↓ | TextVQA Acc. ↑ | TextVQA Exit Layer ↓ | MM-Vet Acc. ↑ | MM-Vet Exit Layer ↓ | SQA-I Acc. ↑ | SQA-I Exit Layer ↓ | Average Acc. ↑ | Average Exit Layer ↓ |
|---|---|---|---|---|---|---|---|---|---|---|---|---|---|
| ✓ | | | | **61.4** | 21.6 | 57.3 | 21.0 | **31.5** | 21.4 | 69.7 | 21.7 | **55.0** | 21.4 |
| | ✓ | | | 61.2 | 21.3 | 57.1 | 21.1 | 30.0 | **21.0** | 69.7 | 21.4 | 54.5 | 21.2 |
| | | ✓ | | 60.1 | 17.4 | **57.6** | 22.1 | 31.1 | 22.1 | 69.7 | 20.4 | 54.6 | 20.5 |
| | | | ✓ | 59.8 | **16.3** | 56.3 | 19.1 | 29.9 | 21.0 | 69.8 | 13.8 | 54.0 | 17.6 |
| ✓ | ✓ | | | **61.1** | 21.2 | 57.3 | 21.3 | 31.6 | 21.3 | 69.7 | 21.0 | **54.9** | 21.2 |
| ✓ | | ✓ | | 58.7 | 16.5 | **57.6** | 22.2 | **32.7** | 22.5 | 69.7 | 21.3 | 54.7 | 20.6 |
| ✓ | | | ✓ | 59.4 | **16.3** | 56.5 | 19.9 | 30.9 | 21.8 | 69.7 | 14.4 | 54.1 | 18.1 |
| | ✓ | ✓ | | 59.4 | 16.7 | 57.4 | 21.8 | 31.3 | 22.0 | 69.6 | 20.3 | 54.4 | 20.2 |
| | ✓ | | ✓ | 59.3 | 16.4 | 56.5 | 19.9 | 31.2 | 21.6 | 69.6 | 14.4 | 54.1 | 18.1 |
| | | ✓ | ✓ | 60.0 | 16.4 | 56.6 | **19.7** | 31.9 | 21.1 | 69.6 | **13.4** | 54.5 | 17.6 |
| | ✓ | ✓ | ✓ | 60.1 | 16.4 | 57.0 | 20.2 | 30.8 | **21.3** | 69.6 | 14.1 | 54.4 | 18.0 |
| ✓ | | ✓ | ✓ | **60.2** | 16.7 | 57.1 | 20.3 | 31.0 | 21.6 | 69.7 | 14.3 | **54.5** | 18.2 |
| ✓ | ✓ | | ✓ | 57.6 | 15.9 | **57.3** | 22.2 | **31.7** | 22.1 | 69.7 | 20.1 | 54.1 | 20.2 |
| ✓ | ✓ | ✓ | | 57.8 | **15.2** | 56.2 | **19.3** | 30.0 | 21.6 | 69.3 | **11.7** | 53.3 | **16.9** |
| ✓ | ✓ | ✓ | ✓ | 60.5 | 16.8 | 57.1 | 20.5 | 31.1 | 21.8 | 69.8 | 14.1 | 54.6 | 18.3 |
| LLaVA-1.5 7B | | | | 62.0 | - | 58.2 | - | 30.5 | - | 69.4 | - | 55.0 | - |

Table 8: Ablation study of the attention status selection for DyVTE. "*Visual Self*" denotes the attention score from visual modality of each visual tokens, similar with "*Text Self*". "*Cross*" denotes the attention score from visual to the text of each visual tokens. For attention representations whose dimensions are not 576, we use interpolation to align the dimensions. "*Exit Layer*" denotes the averaged layer numbers selected to remove by DyVTE. "*Acc.*" denotes the accuracy. The default setting of DyVTE is the last row. The best and second best results are marked in **bold** and underline.

| State | | | GQA | | TextVQA | | MM-Vet | | SQA-I | | Average | |
| Visual Self | Cross | Text Self | Acc. ↑ | Exit Layer ↓ | Acc. ↑ | Exit Layer ↓ | Acc. ↑ | Exit Layer ↓ | Acc. ↑ | Exit Layer ↓ | Acc. ↑ | Exit Layer ↓ |
|---|---|---|---|---|---|---|---|---|---|---|---|---|
| ✓ | | | 38.3 | **0.5** | 42.4 | **0.7** | **12.2** | 0.9 | 64.2 | **0.6** | **39.9** | **0.7** |
| | ✓ | | 38.8 | 1.3 | **43.1** | 1.3 | 11.5 | 1.2 | **65.2** | 1.2 | 39.6 | 1.2 |
| | | ✓ | **39.6** | 1.7 | 42.7 | 1.1 | 11.9 | 3.6 | 64.7 | **0.6** | 39.7 | 1.7 |
| ✓ | ✓ | | 39.4 | **2.7** | **43.1** | 3.0 | 13.1 | **2.6** | 65.4 | 3.8 | 40.3 | 3.0 |
| ✓ | | ✓ | **40.7** | 3.0 | 43.0 | 1.7 | 13.3 | 5.9 | 65.1 | **1.0** | 40.5 | **2.9** |
| | ✓ | ✓ | **40.7** | 3.0 | **43.1** | **1.6** | 13.4 | 6.0 | 65.2 | **1.0** | **40.6** | **2.9** |
| ✓ | ✓ | ✓ | 49.5 | 10.9 | 45.6 | 5.6 | 15.6 | 7.1 | 65.6 | 2.5 | 44.1 | 6.5 |
| LLaVA-1.5 7B | | | 62.0 | - | 58.2 | - | 30.5 | - | 69.4 | - | 55.0 | - |

## A.3 Distribution of exit layers

In Fig. 7, we present the distributions of the visual token exit layers for LLaVA-DyVTE 7B and 13B. From the figure, we observe that DyVTE dynamically selects the exit layer based on the specific task requirements. For example, on the TextVQA dataset, which demands fine-grained information, the exit occurs later compared to other datasets. In contrast, on the GQA dataset, which involves open-ended word questions and simple true/false judgments, the exit layers are more evenly distributed across early and late layers. Another notable observation is that the timing of the exit layer shows similar distributions across MLLMs of different scales and architectures. For instance, both LLaVA-DyVTE 7B and 13B remove all visual tokens at the same layer early in the process on the SQA dataset. Similarly, for InternVL and LLaVA, peaks in the exit layer distributions occur at comparable layers. Overall, these results support our motivation and the proposed DyVTE.

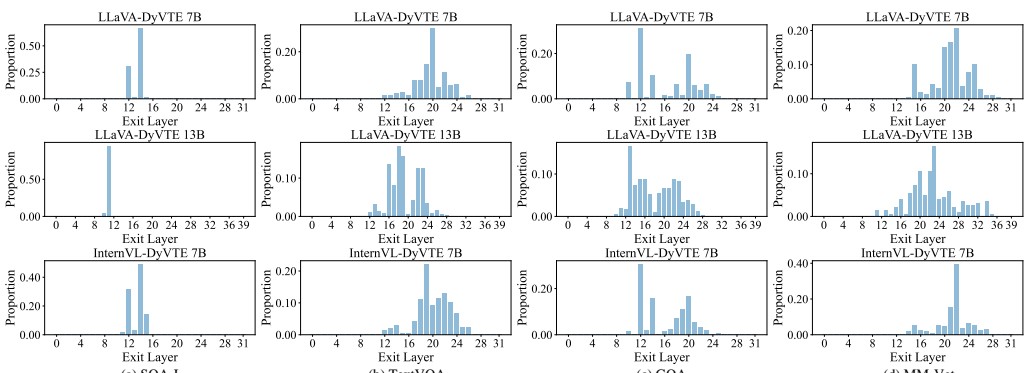

Figure 7: Statistics of the exiting layers decided by DyVTE on LLaVA-1.5 7B, LLaVA-1.5 13B and InternVL 7B. The horizontal axis represents the exit layer, and vertical axis represents the proportion. The exit time by is different for different tasks, but similar for MLLMs with the same sizes.

## A.4 Distribution of Attention

In Fig.8-12, we further visualize the attention distributions on different MLLMs for different datasets. We can first observe that datasets produce similar attention patterns on the same MLLM. Another observation is that attention patterns across different MLLMs have similar trends and the distributions can generally be categorized into three distinct stages. These findings emphasize the universality of attention behavior in MLLMs, further validating our approach. Overall, the experiment results confirm that different MLLMs have similar attention patterns on the different data.

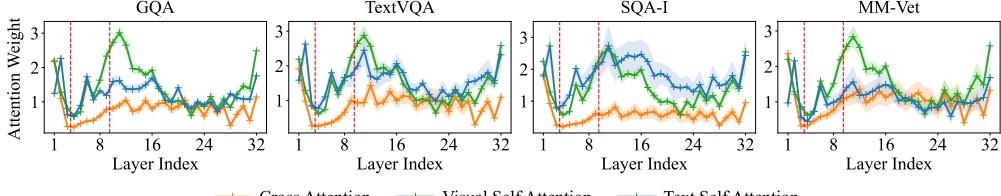

Figure 8: Distributions of averaged attention scores of image self-attention, cross-attention, and text self-attention of LLaVA-1.5 7B. We visualize the mean and variance of the attention weight of each part on GQA, TextVQA, SQA-I, MM-Vet datasets. From these distributions, we can summarize three main stages of MLLMs as introduced in the main paper, which are then marked by red lines.

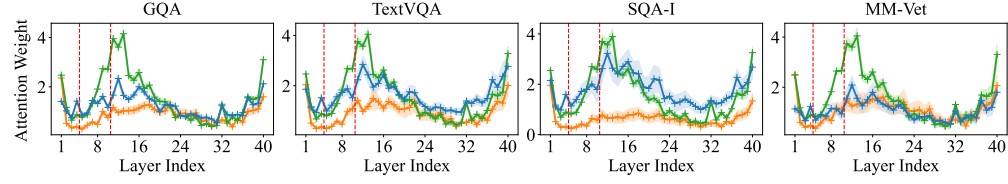

Figure 9: Distributions of averaged attention scores of image self-attention, cross-attention and text self-attention of LLaVA-1.5 13B. We visualize the mean and variance of the attention weight of each part on GQA, TextVQA, SQA-I, MM-Vet datasets. From these distributions, we can summarize three main stages of MLLMs as introduced in the main paper, which are then marked by red lines.

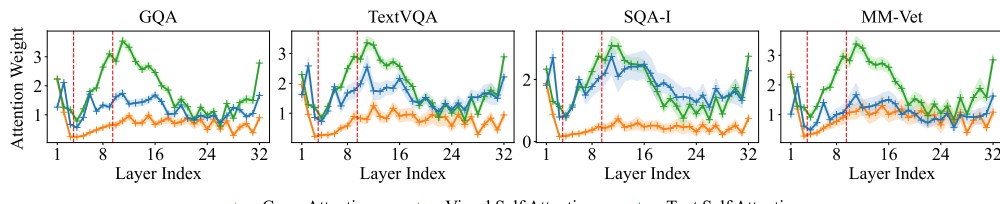

Figure 10: Distributions of averaged attention scores of image self-attention, cross-attention and text self-attention of InternVL 7B. We visualize the mean and variance of the attention weight of each part on GQA, TextVQA, SQA-I, MM-Vet datasets. From these distributions, we can summarize three main stages of MLLMs as introduced in the main paper, which are then marked by red lines.

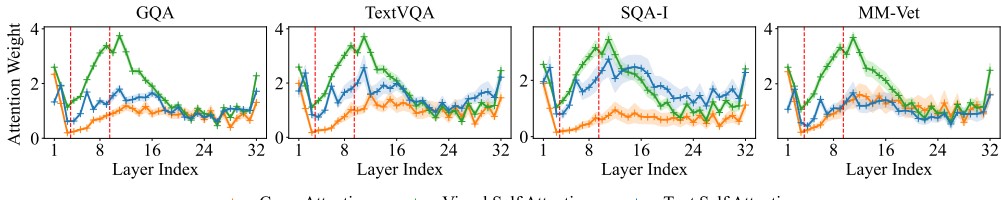

Figure 11: Distributions of averaged attention scores of image self-attention, cross-attention and text self-attention of VILA 7B. We visualize the mean and variance of the attention weight of each part on GQA, TextVQA, SQA-I, MM-Vet datasets. From these distributions, we can summarize three main stages of MLLMs as introduced in the main paper, which are then marked by red lines.

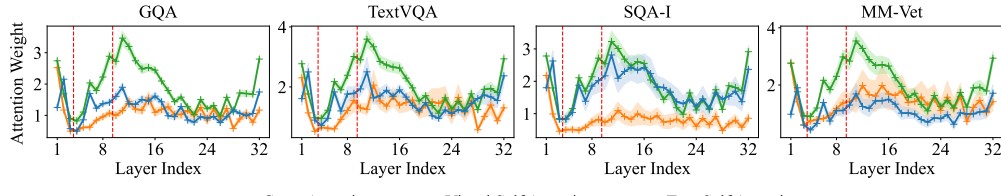

Figure 12: Distributions of averaged attention scores of image self-attention, cross-attention and text self-attention of EAGLE-X5 7B. We visualize the mean and variance of the attention weight of each part on GQA, TextVQA, SQA-I, MM-Vet datasets. From these distributions, we can summarize three main stages of MLLMs as introduced in the main paper, which are then marked by red lines.

## A.5 Entropy of Attention

In Fig.13-17, we visualize the entropy distributions of cross-attention matrices and text self-attention matrices. For all these MLLMs and datasets, we can observe that removing all visual tokens at an appropriate time will not have a significant impact on the answer modeling process. Specifically, for all MLLMs and datasets, the removing of visual tokens only makes the entropy of cross-attention disappear, while the entropy of text self-attention maintains the same distribution. Overall, these results well confirm that cross-modal interaction in fact barely contributes to multimodal reasoning after some layers.

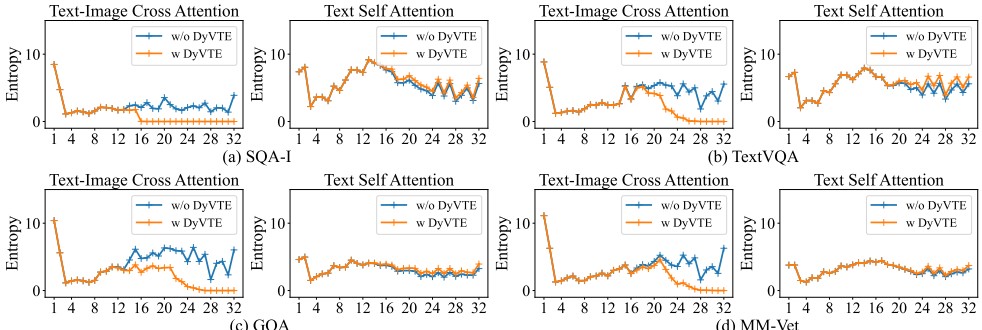

Figure 13: The entropy of two attention distributions. We visualize the entropy of cross-attention matrices and text self-attention on LLaVA-1.5 7B with and without our DyVTE, which shows that the removal of visual tokens barely affect the text self-attention.

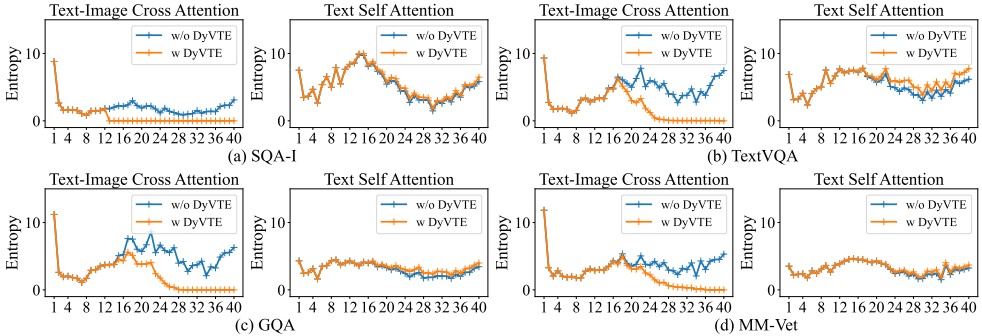

Figure 14: The entropy of two attention distributions. We visualize the entropy of cross-attention matrices and text self-attention on LLaVA-1.5 13B with and without our DyVTE, which shows that the removal of visual tokens barely affect the text self-attention.

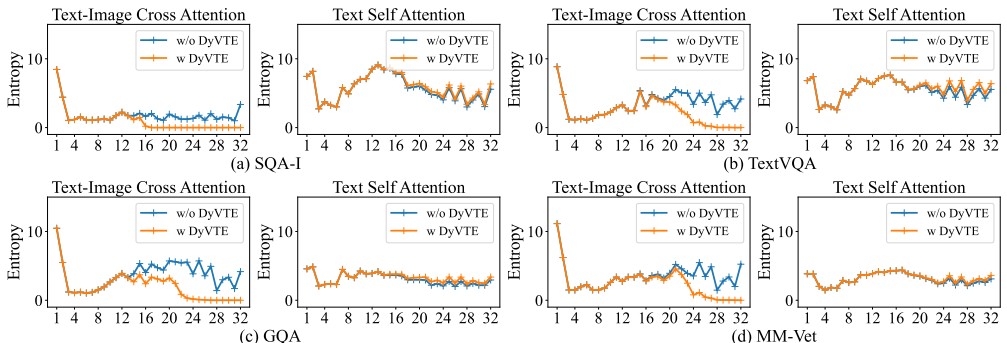

Figure 15: The entropy of two attention distributions. We visualize the entropy of cross-attention matrices and text self-attention on InternVL 7B with and without our DyVTE, which shows that the removal of visual tokens barely affect the text self-attention.

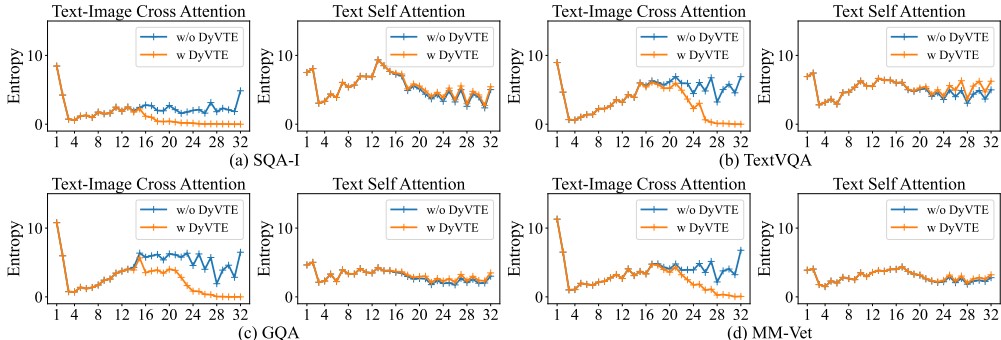

Figure 16: The entropy of two attention distributions. We visualize the entropy of cross-attention matrices and text self-attention on VILA 7B with and without our DyVTE, which shows that the removal of visual tokens barely affect the text self-attention.

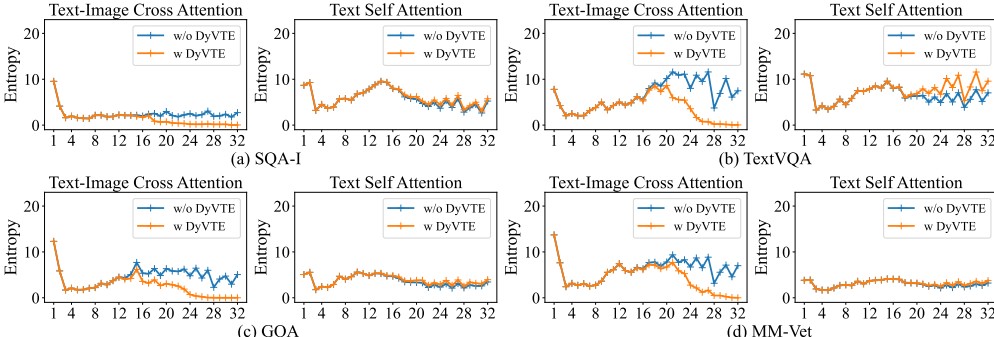

Figure 17: The entropy of two attention distributions. We visualize the entropy of cross-attention matrices and text self-attention on EAGLE-X5 7B with and without our DyVTE, which shows that the removal of visual tokens barely affect the text self-attention.

