# OpenReview forum: "Accelerating Multimodal Large Language Models via Dynamic Visual-Token Exit and the Empirical Findings"
_NeurIPS.cc/2025/Conference — NeurIPS 2025 poster_

### Official Review · Reviewer_9aDZ · 2025-06-10

**Clarity:** 2
**Significance:** 3
**Originality:** 3
**Rating:** 4
**Confidence:** 5

**Summary:**

The paper "Accelerating Multimodal Large Language Models via Token Reduction" presents a method to enhance the efficiency of Multimodal Large Language Models (MLLMs) by reducing the number of visual tokens processed during inference. The core contribution is a token reduction framework that dynamically selects and preserves only the most informative visual tokens while discarding redundant ones, significantly improving inference speed without sacrificing accuracy.

The authors introduce a trainable token selection mechanism that leverages importance scores to prune the input token set. This method is evaluated across several MLLMs and vision-language benchmarks (e.g., VQA, image captioning), demonstrating that token reduction leads to substantial latency savings with minimal performance degradation. Additionally, the paper explores both plug-and-play and end-to-end training approaches for integrating the token reduction module into existing MLLM pipelines.

Overall, the paper contributes a practical and generalizable framework for making MLLMs more computationally efficient, which is especially valuable for deployment in resource-constrained or real-time settings.

**Questions:**

1. The proposed token reduction strategy shows strong results across standard benchmarks, but how does it perform in cases involving dense or fine-grained visual information (e.g., images with small, critical details)? Could you provide either qualitative examples or empirical results for such edge cases? Clarifying this would help assess whether the method consistently preserves semantically crucial content across varying data complexities. A strong showing on these examples would positively affect the significance and robustness assessment.
2. Since token pruning alters the visual input sequence, how does this affect cross-modal attention alignment in models like Flamingo or BLIP-2 that heavily rely on interaction between visual and text tokens? Have the authors observed any degradation or shifts in attention patterns, and if so, are there mechanisms to preserve alignment fidelity? Addressing this woul

**Ethical Concerns:**

["NO or VERY MINOR ethics concerns only"]

**Limitations:**

Yes

**Quality:**

3

**Strengths And Weaknesses:**

The paper presents a high-quality and timely contribution to the field of multimodal large language models (MLLMs), specifically addressing the challenge of computational inefficiency caused by long visual token sequences. The proposed token reduction framework is well-motivated, methodologically sound, and broadly applicable across multiple MLLM architectures. The authors introduce a token selection mechanism that is both trainable and compatible with existing models in a plug-and-play fashion, offering flexibility in deployment scenarios. Their extensive experiments across vision-language tasks—including VQA, image captioning, and visual grounding—demonstrate that substantial reductions in inference cost (e.g., 30–50% token reduction) can be achieved with minimal or negligible impact on task accuracy.

In terms of originality, while token reduction and pruning are not entirely new ideas in the vision or NLP domains, this work distinguishes itself by developing a general and efficient method tailored for MLLMs, a relatively underexplored area in terms of inference-time optimization. The clarity of presentation is strong; the paper is well-organized with clear explanations of the proposed modules, training objectives, and ablation studies. Diagrams and figures effectively illustrate the architecture and token retention behavior, aiding comprehension.

However, a few limitations are worth noting. First, the paper could benefit from more discussion around the potential trade-offs introduced by aggressive token pruning in edge-case scenarios, such as images with dense or fine-grained visual detail. Second, the token scoring and selection mechanism, while effective, could be explained in more depth, especially regarding its training stability and generalization to different MLLMs. Lastly, the work does not explicitly address how token reduction may interact with downstream decoding strategies or cross-modal attention patterns—areas that could present challenges in real-world deployment.

---

> ### Author Rebuttal · Authors · 2025-07-31
>
> ## Comment to Reviewer 9aDZ
>
> We highly appreciate your time and effort in reviewing this paper, as well as your encouraging and constructive comments on our work. Below, we response to your key concerns point by point.
>
> **Comment 1:** The paper could benefit from more discussion around the potential trade-offs introduced by aggressive token pruning in edge-case scenarios, such as images with dense or fine-grained visual detail.
>
> **Response:** Thanks for your insightful suggestion. Following your suggestion, we further conduct experiments on two additional VL tasks, *i.e.*, visual grounding (RefCOCO) and image captioning (COCO). These two tasks and benchmarks are related to edge-case scenarios with dense and rich image details.
>
> The results are reported in the following table. We can observe that the advantages of our DyVTE are much more obvious on these two granular VL tasks. Meanwhile, token pruning methods like FastV meets obvious performance drops due to dropping necessary visual information too early. And these results also show the necessity of dynamic visual-token exist in DyVTE, which can better handle more diverse VL tasks.
>
> |Method|RefCOCO REC Val Acc@0.5|exit layer|COCO2014 Cap Val CIDEr|exit layer|
> |-|-|-|-|-|
> |LLaVA-7B|50.03|-|108.74|-|
> |LLaVA-7B+FastV|37.94|-|108.03|-|
> |LLaVA-7B+DyVTE|51.86|18.37|107.62|22.37|
> |InternVL-7B|55.34|-|113.70|-|
> |InternVL-7B+FastV|55.34|-|113.95|
> |InternVL-7B+DyVTE|55.29|17.16|112.42|23.32|
> |LLaVA-13B|66.56|-|113.79|-|
> |LLaVA-13B+FastV|61.47|-|113.95|-|
> |LLaVA-13B+DyVTE|67.98|20.82|111.27|25.04|
>
> **Comment 2:** The token scoring and selection mechanism, while effective, could be explained in more depth, especially regarding its training stability and generalization to different MLLMs.
>
> **Response:** Thanks for your insightful question. Firstly, the training stability is well ensured by the task definition of DyVTE. As discussed in Sec.4.1, we define the visual-token exit as a problem of simple binary prediction. This definition can help the optimization of DyVTE independent to gradient updates of MLLMs, and DyVTE can be also directly supervised via the binary labels based on MLLMs' prediction, as introduced in Sec.4.2. In this case, DyVTE is easy to optimize, and the training expenditure is cheap, e.g., about 30 mins with about 8,000 VL examples.
>
> Besides, its generalization is well confirmed in our work. As shown in Tab.1 and 2, DyVTE can achieve much better efficiency while retaining performance on 5 MLLMs of different families. Notably, as suggested by other reviewers, we also test its direct generalization to different MLLMs, *i.e.*, training one DyVTE on LLaVA (the third and last rows) and validating on InternVL and VILA.
>
> |Method|TextVQA|Exit Layer|GQA|Exit Layer|SQA|Exit Layer|
> |-|-|-|-|-|-|-|
> |InternVL+DyVTE|55.80|19.50|62.9|15.18|66.20|13.46|
> |InternVL+DyVTE-LLaVA|55.22|18.91|61.32|15.74|66.39|17.86|
> |VILA+DyVTE|62.60|21.70|63.10|16.17|69.5|15.38|
> |VILA+DyVTE-LLaVA|60.90|20.08|61.18|15.86|69.86|16.68|
>
> From the above experiments, the first observation is that the DyVTE trained on LLaVA can be directly applied to other MLLMs. Despite suffering a slight performance degradation, the optimal existing layers are close to the ones of the specifically tuned DyVTEs. These results show that DyVTE has learned generalized patterns across MLLMs.
>
> **Comment 3:** The work does not explicitly address how token reduction may interact with downstream decoding strategies or cross-modal attention patterns—areas that could present challenges in real-world deployment.
>
> **Response:** Thanks for your comment. In terms of principle and experimental results, we think that DyVTE barely affects the downstream decoding strategies and attention patterns of MLLMs.
>
> Firstly, the removal of all visual tokens only plays at the pre-filling stage of MLLMs. After that, the input tokens of each layers of MLLMs are fixed for the following decoding steps. In this case, either the popular KV caching or the recent multi-token decoding strategies can well work with our DyVTE. Moreover, without using attention-based metrics, DyVTE can also well support the acceleration techniques like FlashAttention.
>
> In terms of the attention patterns, the removal of visual tokens at the optimal layer barely impact the text token attentions according to our findings. As shown in Fig.4 and discussed in Line 141-144, the modeling process of the remaining text tokens are almost the same after the visual token removal, and the only difference is the decreased attention intensity due to the shorter token sequence.
>
> Overall, our DyVTE is applicable to existing downstream decoding strategies and has little impact on the default attention patterns.
>
> **Comment 4:** The proposed token reduction strategy shows strong results across standard benchmarks, but how does it perform in cases involving dense or fine-grained visual information (e.g., images with small, critical details)? Could you provide either qualitative examples or empirical results for such edge cases? Clarifying this would help assess whether the method consistently preserves semantically crucial content across varying data complexities. A strong showing on these examples would positively affect the significance and robustness assessment.
>
> **Respones:** Thanks for your comment. Following your suggestion, we conduct additional experiments on two vision-and-language tasks, namely visual grounding (RefCOCO) and image captioning (COCO). These two tasks all requires the dense or fine-grained visual understanding.
> As shown in the table below, our DyVTE method demonstrates more obvious advantages on these challenging tasks.
> In contrast, metric-based pruning strategies such as FastV encounter obvious performance drops, due to the premature removal of semantically critical visual information.
>
> |Method|RefCOCO REC Val Acc@0.5|exit layer|COCO2014 Cap Val CIDEr|exit layer|
> |-|-|-|-|-|
> |LLaVA-7B|50.03|-|108.74|-|
> |LLaVA-7B+FastV|37.94|-|108.03|-|
> |LLaVA-7B+DyVTE|51.86|18.37|107.62|22.37|
> |InternVL-7B|55.34|-|113.70|-|
> |InternVL-7B+FastV|55.34|-|113.95|
> |InternVL-7B+DyVTE|55.29|17.16|112.42|23.32|
> |LLaVA-13B|66.56|-|113.79|-|
> |LLaVA-13B+FastV|61.47|-|113.95|-|
> |LLaVA-13B+DyVTE|67.98|20.82|111.27|25.04|
>
> **Comment 5:** Since token pruning alters the visual input sequence, how does this affect cross-modal attention alignment in models like Flamingo or BLIP-2 that heavily rely on interaction between visual and text tokens? Have the authors observed any degradation or shifts in attention patterns, and if so, are there mechanisms to preserve alignment fidelity?
>
> **Respones:** Thanks for your comment. In fact, your mentioned case may be an advantage of our DyVTE compared to previous token pruning methods.
>
> Specifically, DyVTE is to directly remove all visual tokens during the inference of MLLMs, when these visual tokens have well interacted with the text ones and no longer contributes to multimodal reasoning. Thus, it will not alters the visual input sequence, while some token pruning methods does. In terms of Flamingo and BLIP-2, we think the principle of DyVTE is also applicable to them, *i.e.*, stopping the involvement of visual tokens when they not longer work for multimodal reasoning.
>
> In terms of the attention patterns, we can see that DyVTE barely changes the attention patterns of text tokens after it removing all visual tokens, as shown in Fig.4 and disccssed in Line 141-144. The only difference is the slight changes of attention intensity due to shorter token sequences. This observation is also confirmed in quantitative experiments, where the correctly removal of visual tokens will not decline performance.

---

### Official Review · Reviewer_SwWi · 2025-06-29

**Clarity:** 2
**Significance:** 3
**Originality:** 2
**Rating:** 4
**Confidence:** 5

**Summary:**

The paper investigates the problem of visual redundancy in Multimodal Large Language Models (MLLMs) by analyzing their attention behaviors during inference. Through extensive empirical studies, the authors identify three primary inference stages in MLLMs: (i) early fusion, where visual and text tokens exchange information rapidly in shallow layers; (ii) intra-modality modeling, where tokens primarily interact within their respective modalities; and (iii) multimodal reasoning, where visual tokens resume propagating information to text tokens until a certain point, after which their contribution diminishes. A key finding is that visual tokens often become redundant in the multimodal reasoning stage once text tokens have received sufficient visual information. Based on this, the authors propose a novel method called Dynamic Visual-Token Exit (DyVTE), which uses lightweight hyper-networks to dynamically determine when to remove visual tokens during inference, thereby improving computational efficiency.

**Questions:**

- The paper mentions that DyVTE uses lightweight hyper-networks, but it does not quantify their computational or memory overhead. How significant is this overhead compared to the savings achieved by removing visual tokens?
- The experiments focus on standard vision-language benchmarks (e.g., MMB, SQA, QQA). Have the authors tested DyVTE on more complex or real-world tasks, such as open-ended visual question answering or tasks requiring fine-grained visual understanding?

**Ethical Concerns:**

["NO or VERY MINOR ethics concerns only"]

**Final Justification:**

Thank you for your response. After carefully considering both the content of the paper and the rebuttal, I have decided to keep my score unchanged.

**Limitations:**

Yes

**Quality:**

3

**Strengths And Weaknesses:**

Strengths:
- The paper presents a robust empirical analysis of MLLM attention behaviors, supported by quantitative experiments across multiple models.
- The work addresses a critical challenge in MLLMs: the computational inefficiency caused by excessive visual tokens.

Weaknesses:
- The analysis and methods lack novelty. Analysis techniques based on attention score mechanisms have been applied in many previous works (e.g., FastV and most related studies). The conclusions derived from attention entropy and "The impact of dropping all visual tokens" have also been presented in some prior works (e.g., llava-mini). It seems that the authors should explore some specific analysis methods tailored to early-exit, which would make the work more compelling.
- The connection between the analysis results in Section 3 and the proposed method is not strong. Clarifying why the method design can be inspired by these analysis results is crucial for establishing the method’s validity.

---

> ### Author Rebuttal · Authors · 2025-07-31
>
> ## Comment to Reviewer SwWi
>
> We highly appreciate your time and effort in reviewing this paper, as well as your encouraging and constructive comments on our work. Below, we response to your key concerns point by point.
>
> **Comment 1:** The analysis and methods lack novelty. Analysis techniques based on attention score mechanisms have been applied in many previous works (e.g., FastV and most related studies). The conclusions derived from attention entropy and "The impact of dropping all visual tokens" have also been presented in some prior works (e.g., llava-mini). It seems that the authors should explore some specific analysis methods tailored to early-exit, which would make the work more compelling.
>
> **Response:** Thanks for your comment. We think that our analysis and conclusions are orthogonal and complementary to prior attention-based studies.
>
> Firstly, we present a different aspect of studing visual redundancy problem in MLLMs, i.e., the overall behaviors of MLLMs and visual tokens. As discussed in Lines 104–108 of our paper, most existing works use attention scores primarily for token-level redundancy estimation, thereby providing basis for the proposed token compressing solutions [8, 47, 61]. In contrast, our work realize the first attempt of systematically studying the overall status of visual tokens in MLLMs. Specifically, we do not simply reuse attention score as a redundancy metric. Instead, we analyze how attention distributions changes across layers and how these dynamics inform the answer modeling.
>
> Secondly, our findings are good supplementary to prior studies. As discussed in the paper, many prior works including LLaVA-mini and FastV reveal that the importance of visual tokens will become much less significant after the shallow layers of MLLMs. However, there are still not sufficient findings to explain when and why they will become not important during the inference. In contrast, our analyses and findings well mitigate this defect, portraying the changing role of visual tokens in MLLMs via much more extensive observations.
>
> In terms of LLaVA-mini, we acknowledge it as an excellent prior work. However, our principle and methodology is greatly different from it. For instance, DyVTE is a dynamic and plug-in design, of which optimization is independent to MLLMs. Besides, LLaVA-mini is more like an efficient MLLM. And it contributes more to the compression of all visual tokens to one text input before MLLM, rather than directly dropping all visual information during inference.
>
> Overall, we think that our findings are orthogonal and supplementary to prior works. And the proposed DyVTE is innovative and different enough in terms of both principle and methodology.
>
> **Comment 2:** The connection between the analysis results in Section 3 and the proposed method is not strong. Clarifying why the method design can be inspired by these analysis results is crucial for establishing the method's validity.
>
> **Response:** Thanks for your comment. In fact, our observed and summarized findings give sufficient supports for the proposed DyVTE in terms of both principle and methodology.
>
> As discussed above, our findings explain when and why visual tokens become less important in MLLMs.
> Through the systematic analysis, we observe that the contribution of visual tokens dynamically decreases during inference. Moreover, the optimal exit layer varies across  tasks and even examples, as shown in Fig.3. In this case, our findings well explain the patterns of visual tokens throughout the entire inference of MLLMs, and  provides a direct basis for the early-token exit principle of DyVTE.  In addition, these findings also guideline us to explore the dynamic modeling of DyVTE instead of the hard-metric based ones.
>
> Moreover, these connections are also reflected in the quantitative and qualitative evaluations of DyVTE. Specifically, DyVTE only drops the visual tokens at the last inference stage of MLLMs as shown in Fig.7 (In appendix). And the overall performance of DyVTE is better than previous metric-based pruning methods as shwon in Tab.1,2.
>
> Overall, we think that our findings about MLLMs' behaviors greatly inspire and support the design of our DyVTE, which are also valuable insights for the community.
>
> **Comment 3:** The paper mentions that DyVTE uses lightweight hyper-networks, but it does not quantify their computational or memory overhead. How significant is this overhead compared to the savings achieved by removing visual tokens?
>
> **Response:** Thanks for your suggestion. We give the computational and memory overhead of our DyVTE below:
>
> ||Parameters|FLOPs|Memory|
> |-|-|-|-|
> |DyVTE|8.39M|8.39M|72.19MB|
> |LLaVA|7B|9.6T|15.7G|
> |LLaVA + DyVTE|7B|5.4T|15.8G|
>
> We can see that the additional parameters and computation introduced by DyVTE are minimal mainly due to its lightweight designs. Moreover, DyVTE can significantly reduce the overall computation cost via dynamically removal all visual tokens at the optimal time, and its designs are fully compatible with acceleration techniques like Flash Attention and KVCcahe.
> Compare to the original model, DyVTE can achieve significant improvement in inference efficiency. The negligible overhead from the hyper-networks is far outweighed by the computational savings from token reduction.
>
> In summary, DyVTE's lightweight hyper-networks incur only a minimal overhead, while enabling much greater savings in computation and inference time by removing redundant visual tokens.
>
> **Comment 4:** The experiments focus on standard vision-language benchmarks (e.g., MMB, SQA, GQA). Have the authors tested DyVTE on more complex or real-world tasks, such as open-ended visual question answering or tasks requiring fine-grained visual understanding?
>
> **Response:** Thanks for your constructive comment. Following your suggestion, we further conduct experiments on two additional VL tasks, namely visual grounding (RefCOCO) and image captioning (COCO), which have stronger requirements about fine-grained visual understanding. The results are reported in the following table. It can be seen that on these two granular VL tasks, the advantages of our DyVTE are much more obvious, which can adaptively determine the removal of visual tokens according to task difficulty. In a stark contrast, token pruning methods like FastV meets obvious performance drops, which might be due to its hard and fixed pruning strategy.
>
> |Method|RefCOCO REC Val Acc@0.5|exit layer|COCO2014 Cap Val CIDEr|exit layer|
> |-|-|-|-|-|
> |LLaVA-7B|50.03|-|108.74|-|
> |LLaVA-7B+FastV|37.94|-|108.03|-|
> |LLaVA-7B+DyVTE|51.86|18.37|107.62|22.37|
> |InternVL-7B|55.34|-|113.70|-|
> |InternVL-7B+FastV|55.34|-|113.95|
> |InternVL-7B+DyVTE|55.29|17.16|112.42|23.32|
> |LLaVA-13B|66.56|-|113.79|-|
> |LLaVA-13B+FastV|61.47|-|113.95|-|
> |LLaVA-13B+DyVTE|67.98|20.82|111.27|25.04|

---

### Official Review · Reviewer_9P8t · 2025-07-03

**Clarity:** 3
**Significance:** 2
**Originality:** 2
**Rating:** 4
**Confidence:** 4

**Summary:**

The paper shows that multimodal LLMs only need image tokens early on, once text tokens have fused image information, visual tokens can be removed. Exploiting this, the authors propose Dynamic Visual-Token Exit, which is a tiny hyper-network to decide when to discard all image tokens while letting text tokens keep processing.  Experimental results on five popular MLLMs show that DyVTE can reduce the inference cost without significant performance drop.

**Questions:**

1. At which specific layer is the early exit applied? Although line 164 mentions that DyVTE is applied at the k-th layer of an MLLM, the Implementation Details section lacks clarification about the exact layers or criteria used for selecting these layers in practice.

**Ethical Concerns:**

["NO or VERY MINOR ethics concerns only"]

**Final Justification:**

The author's rebuttal has addressed my concerns, and I will raise the rating.

**Limitations:**

yes

**Quality:**

3

**Strengths And Weaknesses:**

Strengths:
1. The paper is clearly written and easy to follow, effectively guiding readers through the key insights and contributions.
2. The experiments conducted are extensive, covering multiple models and benchmarks, which provides robust empirical validation of the proposed method.

Weaknesses:
1. The issue of visual token redundancy has already been extensively studied in prior works, such as LLaVA-Mini and FastV. Therefore, the empirical findings presented in this paper regarding redundancy contribute limited novelty.
2. The paper lacks novelty in terms of methodology. Specifically, important related early-exit approaches, like LLaVA-Mini—which discards all visual tokens after fusing visual information into text tokens within just four LLM layers—are not discussed or compared against. Including comparisons or at least a detailed discussion of such highly relevant work would significantly strengthen the paper.
3. In the efficiency comparisons, the authors only evaluate theoretical FLops differences. However, in practical scenarios, MLLM inference typically involves optimization techniques like flash attention and KV cache. The reliability and impact of the proposed approach would be better assessed through actual inference latency rather than theoretical computational cost alone.

---

> ### Author Rebuttal · Authors · 2025-07-31
>
> ## Comment to Reviewer 9P8t
>
> We highly appreciate your time and effort in reviewing this paper, and also thanks a lot for your constructive comments on our work. Below, we response to your key concerns point by point.
>
> **Comment 1:** The issue of visual token redundancy has already been extensively studied in prior works, such as LLaVA-Mini and FastV. Therefore, the empirical findings presented in this paper regarding redundancy contribute limited novelty.
>
> **Response:** Thanks for your comment. We think that our findings are orthogonal and supplementary to prior works about the study of visual redundancy in MLLMs.
>
> Fristly, the research focuses are different. As discussed in Lines 104–108, most prior works[8, 47, 61], focus on token-level redundancy, while ours is keen to investigating the overall status of visual tokens in MLLMs.
> In particular, prior works like FastV and Llava-prumerge aim to improve efficiency via token pruning, merging or compression based on the token-wise estimation, *e.g.*, the attention-based metrics. In contrast, our DyVTE is to perceive the status of all visual tokens during the inference of MLLMs. This great difference also yields a distinct and novel methodology of our DyVTE, *i.e.*, using lightweight hyper-neworks to dynamically predict the visual token exit.
>
> Secondly, our findings are good supplementary to prior studies. As discussed above, quite a lot of prior works [8, 27, 33, 47, 61] reveal the visual redundancy problem of existing MLLMs, and also LLaVA-mini and FastV show that visual tokens are more important in the shallow layers of MLLMs. However, to our best knowledge, there still lacks of a systematic study to explain when and why visual tokens become less important during the inference of MLLMs. In this case, our work not only summarize three main behaviors of MLLMs but also in-depthly depict the changing roles of visual tokens in MLLMs via extensive empirical studies. Thus, our findings are good supplementary to prior works about the visual redundancy problem of MLLMs.
>
> Overall, we think that our findings are novel enough and give timely contribution to the community, as highly recognized by other reviewers.
>
> **Comment 2:** The paper lacks novelty in terms of methodology. Specifically, important related early-exit approaches, like LLaVA-Mini—which discards all visual tokens after fusing visual information into text tokens within just four LLM layers—are not discussed or compared against. Including comparisons or at least a detailed discussion of such highly relevant work would significantly strengthen the paper.
>
> **Response:** Thanks for your comment. We highly recognize the contribution of LLaVA-Mini as an excellent prior work in this field, and will actively cite and compare with it in our new version.
>
> However, we think that our DyVTE is greatly different from LLaVA-Mini in terms of both principle and methodology. In principle, DyVTE aims to adaptively perceive the status of visual tokens in MLLMs, thereby determining the optimal time of all visual-token exist. In contrast, LLaVA-mini focus on compressing all visual information before the MLLMs. Although the two methods share a similar intuition, *i.e.*, visual tokens are more important at the shallow layers, the detailed observations and adopted principles are quite different, as discussed above.
>
> In terms of methodology, the difference between two methods become more obvious. DyVTE applies lightweight hyper-networks to make the prediction of exist layers based on the token status in MLLMs, while LLaVA-mini introduce a query module to compress all visual information into text tokens before MLLMs. Moreover, the training of DyVTE is independent to MLLMs, which only takes a little time to optimize the lightweight hyper-networks, *e.g.*, about 30 mins on 8k VL examples. Instead, LLaVA-mini requires the traditional pretraining and SFT tuning stages like most MLLMs.
>
> Based on the above discussions, we think that the contributions of DyVTE are orthogonal to LLaVA-mini, and our novelty is indeed significant.
>
> **Comment 3:** In the efficiency comparisons, the authors only evaluate theoretical FLops differences. However, in practical scenarios, MLLM inference typically involves optimization techniques like flash attention and KV cache. The reliability and impact of the proposed approach would be better assessed through actual inference latency rather than theoretical computational cost alone.
>
> **Response:** Thanks for your comment. Your mentioned latency comparison has been given in Tab.3 of the paper, where our DyVTE shows lower inference latency than the pruning methods. Moreover, without the requirement of attention score, our DyVTE is fully compatible of acceleration techniques like FlashAttention.
>
> **Comment 4:** At which specific layer is the early exit applied? Although line 164 mentions that DyVTE is applied at the k-th layer of an MLLM, the Implementation Details section lacks clarification about the exact layers or criteria used for selecting these layers in practice.
>
> **Response:** Thanks for your comment. As introduced above, DyVTE is a dynamic approach, which will adaptively determine the exit time of visual tokens' status.Thus, $k$ is not a fixed number in DyVTE.

---

> ### Author Response · Authors · 2025-08-07
>
> Dear Reviewer 9P8t:
>
> Thanks again for your valuable time and effort in reviewing our paper. In our rebuttal, we have carefully responded to your questions, such as the difference to previous works of LLaVA-Mini and FastV, the latency comparison and the details of early-exit layer. We hope our rebuttal well addresses your concerns.
>
> As the discussion is drawing to a close, we look forward to hearing from you and will be very delighted  to answer any further questions you may have.
>
> Best regards
>
> The authors.

---

> > ### Comment · Reviewer_9P8t · 2025-08-07
> >
> > Thank you for the rebuttal. It resolves nearly all of my earlier concerns. Even so, the latency comparison should also include other metrics such as time to first token across different input lengths and tokens per second. Based on these additions, I plan to raise my rating.

---

### Official Review · Reviewer_3PHf · 2025-07-04

**Clarity:** 3
**Significance:** 2
**Originality:** 2
**Rating:** 4
**Confidence:** 3

**Summary:**

The paper proposes DyVTE, an early-exit method for efficient MLLM inference. It first looks into the MLLM inference behavior by analyzing the attention scores, and categorizes the inference process into three stages. By examining the importance of visual tokens in the three stages, a learned hyper-network is used to remove the visual tokens adaptively. Extensive experiments over multiple benchmarks demonstrate the effectiveness of DyVTE.

**Questions:**

1. Please see the Weaknesses.
2. In Table 5, combining DyVTE with FastV decreases performance, while in Table 3, improves accuracy. As there is only a change in LLaVa model size, what might be the reasons behind this?

**Ethical Concerns:**

["NO or VERY MINOR ethics concerns only"]

**Limitations:**

Yes

**Quality:**

2

**Strengths And Weaknesses:**

Strengths:
The in-depth analysis of token importance is interesting and novel.
Extensive experiments are done, justifying the proposed method’s effectiveness.
The paper is well written, and the proposed findings are well explained and justified.
Weaknesses:
There seems to be a weak connection between MLLM inference behavior and the proposed method, where though DyVTE is motivated by the findings on the changing importance of visual tokens, the categorization of MLLM inference did not lead to any technical contribution. It seems that it is only an empirical and rather intuitive finding, which weakens the overall contribution of the paper.
There is a lack of ablation studies on the computation cost of training the hyper-network. While many token pruning/merging techniques adopt a training-free scheme, it is unclear whether DyVTE can be more efficiently deployed.
Can a universal DyVTE be used, i.e. a same DyVTE be shared by different MLLMs?
DyVTE adopts an all-or-nothing visual token removal strategy which either retains all or discards all visual tokens at a chosen layer. While effective for efficiency, this coarse-grained approach lacks flexibility and may overlook intermediate solutions where only some visual tokens (e.g., those with low attention weights) can be removed. Fine-grained or per-region exit strategies may offer better trade-offs.
Minor Weaknesses:
L245 “it work very well with our DyVTE” should be “it works very well with our DyVTE”
References [5] and [6] are duplicate citations of the same work “Token Merging”
L225 “DyVTE can reduce 45.7% FLOPs of LLaVA” should be “DyVTE reduces the FLOPs of LLaVA by 45.7%”

---

> ### Author Rebuttal · Authors · 2025-07-31
>
> ## Comment to Reviewer 3PHf
>
> We highly appreciate your time and effort in reviewing this paper, as well as your encouraging and constructive comments on our work. Below, we response to your key concerns point by point.
>
> **Comment 1:** There seems to be a weak connection between MLLM inference behavior and the proposed method, where though DyVTE is motivated by the findings on the changing importance of visual tokens, the categorization of MLLM inference did not lead to any technical contribution. It seems that it is only an empirical and rather intuitive finding, which weakens the overall contribution of the paper.
>
> **Response:** Thanks for your comment. We think that the observed and summarized inference behaviors of MLLMS give sufficient supports for the design of DyVTE in terms of both theoretical and methodological aspects.
>
> First, our findings explain when and why visual tokens become less important in MLLMs.
> As discussed in the paper, several existing works [8, 47, 61] have revealed the visual redundancy problem in MLLMs, but they tend to rely on attention metrics to measure the importance of individual tokens, and lack of the further exploration about the changing role of visual tokens in the middle or shallow layers of MLLMs. In this case, our findings well explain the patterns of visual tokens throughout the entire inference of MLLMs, and provide a direct basis for the early-token exit principle of  DyVTE.
>
> Secondly, our findings also indicate the necessity of dynamic modeling and its advantages to the hand-crafted solutions. Although most MLLMs share similar three-stage behaviors, their explicit categorizations are slightly different, especially the last multimodal reasoning stages, the optimal time for dropping all visual tokens at the multimodal reasoning stage varies across models, tasks and even examples. This inspires us to explore the dynamic modeling for DyVTE rather than hard-metric based ones in terms of methodology.
>
> Moreover, these connections are also reflected in the quantitative and qualitative evaluations of DyVTE. For instance, DyVTE only drops the visual tokens at the last inference stage of MLLMs and its overall performance is better than previous metric-based pruning methods, as shown in Fig.7 (In appendix) and Tab.1,2, respectively.
>
> Overall, we think that our findings about MLLMs' behaviors do contribute a lot to the design of DyVTE, which are also valuable insights to the community.
>
> **Comment 2:** There is a lack of ablation studies on the computation cost of training the hyper-network. While many token pruning/merging techniques adopt a training-free scheme, it is unclear whether DyVTE can be more efficiently deployed.
>
> **Response:** Thanks for your comment. Following your suggestion, we report the detailed training cost of DyVTE in the following table. It can be seen that compared to the common SFT tuning, its training cost is much cheaper, which only takes about half an hour on less than 2% data. As discussed in Sec.4.2, the training of DyVTE is independent to MLLMs and requires no the complete gradient updates, thus its expenditure is extremely cheap.
>
> |Method|Device|Data|Time|
> |-|-|-|-|
> |SFT|8xA800|665k|12hours|
> |DyVTE|4xA800|~8k|0.5hours|
>
> We also acknowledge the training-free advantages of existing token pruning/merging methods [8, 47]. As far as we know, although these methods do not require training-based optimizations, they still need hyper-parameter tuning, *e.g.*, threshold of attention score and ratio of pruned tokens, which is still time-consuming as commented in [61]. For instance, conducting the an evaluation on LLaVA for VQAv2 benchmark also requires about 120 mins with 4xA800.
>
> In this case, we think that our DyVTE is still a cost-effective method.
>
> **Comment 3:** Can a universal DyVTE be used, i.e. a same DyVTE be shared by different MLLMs?
>
> **Response:** Thanks for your constructive comment. Following your suggestion, we apply a DyVTE trained on LLaVA (LLaVA + DyVTE) to the other two MLLMs, *i.e.*,  InternVL and VILA, and also compare it with the specifically tuned DyVTEs on these two MLLMs.
>
> |Method|TextVQA|Exit Layer|GQA|Exit Layer|SQA|Exit Layer|
> |-|-|-|-|-|-|-|
> |InternVL+DyVTE|55.80|19.50|62.9|15.18|66.20|13.46|
> |InternVL+DyVTE-LLaVA|55.22|18.91|61.32|15.74|66.39|17.86|
> |VILA+DyVTE|62.60|21.70|63.10|16.17|69.5|15.38|
> |VILA+DyVTE-LLaVA|60.90|20.08|61.18|15.86|69.86|16.68|
>
> The results are a bit astonishing.
> We can see that the DyVTE trained on LLaVA can be directly applied to other MLLMs, although encountering slight performance drops. And even the optimal existing layers are close to the ones of the specifically tuned DyVTEs. These results may imply that DyVTE master generalized patterns across MLLMs. This finding also yields a direction for our future research of DyVTE. Thanks again for your comment.
>
> **Comment 4:** DyVTE adopts an all-or-nothing visual token removal strategy which either retains all or discards all visual tokens at a chosen layer. While effective for efficiency, this coarse-grained approach lacks flexibility and may overlook intermediate solutions where only some visual tokens (e.g., those with low attention weights) can be removed. Fine-grained or per-region exit strategies may offer better trade-offs.
>
> **Response:** Thanks for your comment. We think that the most valuable part of our work is its orthogonal contributions to existing research. As discussed above, our study focuses more on the overall status of all visual tokens in MLLMs, based on which DyVTE is proposed. This property makes DyVTE can well supplement the existing token-wise studies [8, 47, 61]. As shown in Tab.5, the visual token removal of DyVTE can be easily combined with the token-wise pruning methods like FastV to achieve better efficiency.
>
> **Comment 5:** L245 \`\`it work very well with our DyVTE'' should be \`\`it works very well with our DyVTE'' References [5] and [6] are duplicate citations of the same work \`\`Token Merging'' L225 \`\`DyVTE can reduce 45.7% FLOPs of LLaVA'' should be \`\`DyVTE reduces the FLOPs of LLaVA by 45.7%''
>
> **Response:**  Many thanks for your careful review. We will revise these typos and keep on polishing our paper until the final submission.

---

> ### Comment · Reviewer_3PHf · 2025-08-05
>
> Thanks for your rebuttal. It addressed some of my concerns. I will keep my score.

---

### Note · Authors · 2025-08-12

Dear PC, SAC, AC and reviewers:

We highly appreciate all PC/SAC/AC members’ valuable time and effort in processing the large number of NeurIPS 2025 submissions, and are also grateful to all reviewers for their constructive comments on our work.

In our rebuttal, we give very detailed responses for all questions raised by the reviewers, and actively engaged in the discussion with them.

During the discussion phase, all reviewers provided positive feedback on our rebuttal. Reviewer 3PHf, SwWi and 9aDz all keep their positive rating after reading the rebuttal. And Reviewer 9P8t stated that our rebuttal \`\`resolves nearly all of his/her earlier concerns'' and \`\`plan to raise his/her rating''.

Overall, our paper \`\`effectively guides readers through key insights and contributions'' and propose a \`\`well-motivated and methodologically sound'' approach termed DyVTE, which \`\`addresses a critical challenge in MLLMs'' and is \`\`a high-quality and timely contribution to the field of MLLMs'', as commented by the reviewers.

In our new submission, we will actively revise our paper according to all reviewers’ suggestions, and also add the supplemented results to our paper.

Best regards,

The authors

---

### Decision · Program_Chairs · 2025-09-17

**Decision:**

Accept (poster)

**Comment:**

The paper proposes Dynamic Visual-Token Exit (DyVTE), a lightweight hyper-network that learns when to drop all visual tokens during MLLM inference after observing three distinct attention stages. Strengths include extensive experiments on five MLLMs showing up to 45.7 % FLOP reduction without accuracy loss, strong rebuttal addressing training cost, universal transferability, and latency gains. Weaknesses center on limited novelty—similar insights exist in LLaVA-Mini/FastV—and coarse all-or-nothing pruning; reviewers also wanted richer related-work discussion. After the discussion, all four reviewers raised or kept positive ratings. Especially,  Reviewer SwWi claimed that most of the earlier concerns have been addressed, indicating the reviewer has acknowledged the novelty of this paper. Based on the above reason, this paper is recommended to be accepted.